

# Phylogenetic revision of the psammophilic *Trogloderus* LeConte (Coleoptera: Tenebrionidae), with biogeographic implications for the Intermountain Region

M. Andrew Johnston

Biodiversity Knowledge Integration Center, Arizona State University, Tempe, AZ, USA

## ABSTRACT

The genus *Trogloderus* LeConte, 1879, which is restricted to dunes and sandy habitats in the western United States, is revised using morphological and molecular information. Six new species are described from desert regions: *Trogloderus arcanus* New Species (Lahontan Trough); *Trogloderus kandai* New Species (Owens Valley); *Trogloderus major* New Species (Mohave Desert); *Trogloderus skillmani* New Species (eastern Great Basin and Mohave Desert); *Trogloderus verpus* New Species (eastern Colorado Plateau); and *Trogloderus warneri* New Species (western Colorado Plateau). A molecular phylogeny is presented for the genus and used to infer its historical biogeography. The most recent common ancestor of *Trogloderus* is dated to 5.2 mya and is inferred to have inhabited the Colorado Plateau. Current species most likely arose during the mid-Pleistocene where the geographic features of the Lahontan Trough, Bouse Embayment and Kaibab Plateau were significant factors driving speciation.

## INTRODUCTION

The psammophilic genus *Trogloderus* LeConte, 1879, was originally erected for a unique species and specimen of the family Tenebrionidae (sensu *Bouchard et al., 2011*; *Bousquet et al., 2018*). Described as *Trogloderus costatus* LeConte, 1879, from Rock Creek, Idaho, this heavily sculptured species was thought to be similar to the old-world Scaurini Billberg, 1820, but has long since been associated with the desert stink beetles in the genus *Eleodes* Eschscholtz, 1829 in what is now considered the tribe Amphidorini LeConte, 1862 (*LeConte, 1879*; *Blaisdell, 1909*; *Doyen & Lawrence, 1979*; *Bousquet et al., 2018*). *Blaisdell (1909)* described a second congeneric species, *Trogloderus tuberculatus* Blaisdell, 1909, from Los Angeles County, California during his revision of the tribe. A third species, *T. nevadus* La Rivers, 1943, was described from the dunes around Pyramid Lake, Nevada (*La Rivers, 1943*). The genus was then revised by *La Rivers (1946)*, where the three previously recognized species were sunk to subspecies of an again monotypic genus and a fourth subspecies, *T. costatus vandykei* La Rivers, 1946, was described from outside 29 Palms, California.

Corresponding author
M. Andrew Johnston,
ajohnston@asu.edu

The recognition of subspecies (*La Rivers, 1946*) was supported by invoking the theory of orthogenesis, a teleological view of evolution where species have an internal mutational force that drives them not only to a point of adaptation but then continues to push the species onward toward extinction (*Eimer, 1898*; *Mayr, 1982*; see also *Grehan & Ainsworth, 1985*). Following this reasoning, it was hypothesized that *Trogloderus* has "embarked on that phase of evolutionary growth which seems to characterize any ancient group in the last stages of its existence—they are developing fluidly and rapidly into grotesque caricatures of their plain and drab ancestors" (*La Rivers, 1946*: 35).

Following the 1946 revision, very little systematic research has been dedicated to this genus; except for two additional subspecies described as *T. costatus pappi* Kulzer, 1960, and *T. costatus mayhewi* Papp, 1961. All species and subspecies were described from a small number of specimens, with *T. nevadus* having the largest type series of 14 individuals. Subsequent to the above works, specimens in natural history collections have variously been determined as simply *T. costatus* or somewhat haphazardly assigned to subspecies. The last taxonomic changes to the genus were made by this author (MAJ) in the recent catalog of North American Tenebrionidae to stabilize the nomenclature in anticipation of this revision; namely, the subspecific names were all eliminated while restoring *T. costatus*, *T. tuberculatus*, *T. nevadus* and *T. vandykei* to specific standing, while *T. costatus mayhewi* (= *T. vandykei*) and *T. costatus pappi* (= *T. tuberculatus*) were invalidated as junior synonyms (*Bousquet et al., 2018*).

During the half century since the last taxonomic works were completed, a comparatively large number of *Trogloderus* specimens have accumulated in North American natural history collections. These, along with targeted fieldwork for molecular vouchers, have made a thorough taxonomic and biogeographic study of *Trogloderus* possible for the first time.

*Trogloderus* is distributed throughout the Intermountain Region, which encompasses the generally arid lands of western North America between the Rocky and Sierra Nevada mountains. This region spans the Great Basin and Mojave deserts to the west along with the Colorado Plateau to the east. The most comprehensive biogeographic work on the region was completed by *Reveal (1979)*, based largely on his extensive botanical fieldwork. The vast landscape with limited access, particularly in the state of Nevada, has resulted in a general paucity of distributional knowledge and available specimens of beetles in natural history collections (*Will, Madan & Hsu, 2017*).

The molecular phylogenies inferred for the herein revised species-level entities are used in phylogenetic dating analyses to infer the age of the genus and constituent species. They are further used in historical biogeographic reconstructions to understand the geographic influence of the Intermountain Region during speciation. The biogeographic hypotheses generated from these investigations are discussed in relation to other regional treatments. It is hoped that these insights will spur additional studies within the region and provide a framework to understand the relationships between organisms occurring in sand dunes.

## MATERIALS AND METHODS

### Morphological methods

A total of 3,734 specimens were studied. Remarkably, over half (1,957) came from non-institutionalized collections, which is a testament to the importance of individual collections and collectors for documenting North American darkling beetle diversity. The following collections were used for this study:

**ADSC** Aaron D. Smith Collection, Flagstaff, AZ

**AMNH** American Museum of Natural History, New York, NY

**ASUHIC** Hasbrouck Insect Collection, Arizona State University, Tempe, AZ

**CASC** California Academy of Sciences, San Francisco, CA

**CSCA** California State Collection of Arthropods, Sacramento, CA

**CIDA** Orma J. Smith Museum of Natural History, College of Idaho, Caldwell, ID

**EMEC** Essig Museum of Entomology, University of California, Berkeley, CA

**FSCA** Florida State Collection of Arthropods, Gainesville, FL

**FWSC** Frederick W. Skillman Collection, Pearce, AZ

**KKIC** Kojun Kanda Insect Collection, Flagstaff, AZ

**LACM** Natural History Museum of Los Angeles County, Los Angeles, CA

**MAJC** M. Andrew Johnston Collection, Tempe, AZ

**OSUC** Triplehorn Insect Collection, The Ohio State University, Columbus, OH

**RLAC** Rolf L. Aalbu Collection, Sacramento, CA

**SWC** Samuel Wells Collection, Cedar City, UT

**UCDC** Bohart Museum of Entomology,University of California, Davis, Davis, CA

**USNM** National Museum of Natural History, Washington, DC

**WBWC** William B. Warner Collection, Chandler, AZ.

Specimens were examined using a Leica MZ16 stereomicroscope fitted with an ocular graticule for measurements. Internal anatomy was studied via 16 whole-body disarticulations where specimens were cleared in warm 10% KOH, neutralized in 5% acetic acid, and then separated into constituent sclerotized sections in glycerin. Beetle terminalia were further studied from many more specimens by dry dissection. This technique involved prying abdominal ventrites 4–5 and associated internal structures from the pinned specimens, soaking them in distilled water, and separating out the sclerotized reproductive structures. These structures (ovipositor or aedeagus) were then pointed along with the dismembered ventrites and surviving tergites underneath the original pinned specimen.

Morphological terminology generally follows *Doyen (1966)*. Female terminalia morphology follows *Iwan & Kamiński (2016)*, whereas male terminalia morphology follows *Iwan (2001)* except for the usage of the term clavae (following *Blaisdell, 1909*) over laciniae for the ventral articulated structures of the fused parameres that flank the penis. A detailed internal and external description is provided for the genus and each

 

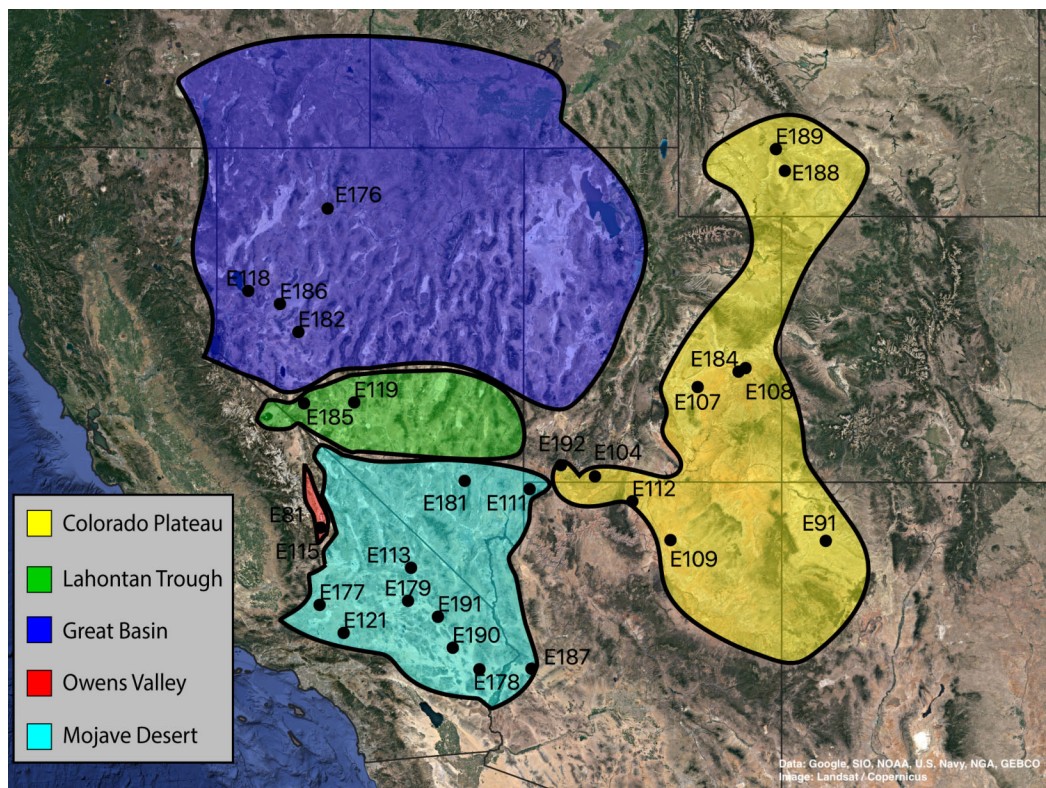

**Figure 1 Collection localities of *Trogloderus* molecular vouchers and biogeographic regions.** Map created in qgis utilizing Google Earth imagery. Map data ©2019 Google, SIO, NOAA, U.S. Navy, NGA, GEBCO.

species is then accompanied by a smaller differential description for the limited variable characters between species.

The evolutionary species concept of *Wiley & Mayden (2000)* is employed in this study. Unique combinations of morphological characters, diagnosable monophyletic clades, and coherent geographic distributions were evaluated together to diagnose putative lineages with a shared evolutionary past and unique evolutionary trajectory.

## Molecular and biogeographic methods

A total of 36 specimens of *Trogloderus* and an additional eight outgroup species from Amphidorini are included in the final matrix. For *Trogloderus*, all type localities were visited, and specimens representing each described species and subspecies were collected. Specimens were collected from as many known localities as possible, with sampling covering all broadly recognized geographic subregions. The collecting locality of each voucher is shown in Fig. 1, and the voucher numbers are included in all presented phylogenetic trees. An additional eight outgroups were included which span the known generic and subgeneric diversity of Amphidorini (*Bousquet et al., 2018*).

Fresh specimens were collected and preserved in 95% ethanol at −20 °C. DNA extractions were made from either the head capsule or a leg and associated thoracic musculature using the DNEasy Blood & Tissue Kit (QIAGEN, Hilden, Germany,

**Table 1 Loci and associated primers used in this study.**

| Locus | Alignment length (bp) | Primers used | Primer source |
|---|---|---|---|
| Cytochrome c oxidase subunit 1 (COI) | 792 | Jerry (F) | *Simon et al. (1994)* |
| | | Pat (R) | |
| Cytochrome c oxidase subunit 2 (COII) | 700 | F-lue (F) | *Whiting (2002)* |
| | | 9b (R) | |
| 12S mitochondrial ribosomal RNA (12S) | 350 | SR-J-14233 (F) | *Simon et al. (1994)* |
| | | SR-N-14588 (R) | |
| 28S ribosomal RNA (28S) | 1,030 | NLF184 (F) | *Van der Auwera, Chapelle & De Wächter (1994)* |
| | | D3ar (R) | *Maddison (2008)* |
| Histone 3 (H3) | 361 | Haf (F) | *Colgan et al. (1998)* |
| | | Har (R) | |
| Wingless (wnt) | 474 | wg550f (F) | *Wild & Maddison (2008)* |
| | | wfAbrZ (R) | |

www.qiagen.com). The six loci amplified via PCR for this study are given in Table 1, PCR cycles and annealing temperatures generally follow *Kanda (2017)*. Forward and reverse sequences were obtained for each PCR product using an Applied Biosystems 3730 DNA Analyzer. The resultant chromatograms were edited for final base calls using Geneious v7 and aligned using MAFFT v7 (*Katoh & Standley, 2013*) as implemented through Mesquite (*Maddison & Maddison, 2018*). The final aligned dataset contained 3,707 base pairs.

All loci were separated into three possible partitions by codon position, except for the ribosomal 12s and 28s which were each left as a single partition, and were analyzed by PartitionFinder 2 (*Lanfear et al., 2016*) using unlinked branch lengths and the greedy search algorithm (*Lanfear et al., 2012*). The resultant two-partition scheme, which consisted of one partition including the third codon position of COI and COII and a second partition containing the rest of the data, was used in downstream phylogenetic and dating analyses. Phylogenetic reconstruction was performed both by RAxML version 8 (*Stamatakis, 2014*) with support values calculated by rapid bootstrap analysis with 500 replicates, and by MrBayes v3.2 (*Ronquist & Huelsenbeck, 2003*) which was run using four chains for 10 million generations sampled every 1,000 with the first 25% being discarded as burnin. Trees were rooted by using the clade containing the three *Eleodes* subgenera *Eleodes*, *Metablapylis* Blaisdell, 1909, and *Steneleodes* Blaisdell 1909 based on previous phylogenetic analyses for the whole tribe (*Johnston, 2018*).

Phylogenetic dating analyses were performed using two methods. First, RelTime (*Tamura et al., 2012*) as implemented in MEGA7 (*Kumar, Stecher & Tamura, 2016*) was used to infer a timetree given the maximum-likelihood tree from RAxML and the aligned nucleotide data. Second, the BEAST2 package (*Bouckaert et al., 2014*) was used to infer a dated phylogeny under both a Yule and Birth-Death model. The latter two analyses had unlinked exponential relaxed clocks for each partition and were run for

500 million generations and sampled every 20,000 with parameter convergence and estimated sample size being assessed via Tracer 1.7 (*Rambaut et al., 2018*) and a maximum clade credibility tree being computed by TreeAnnotator from the BEAST2 package with the first 25% of trees being discarded as burnin.

Two geological calibration points were used for all phylogenetic dating analyses, due to the lack of any fossils for the tribe (*Bousquet et al., 2018*). The first calibration is the uplift of the Inyo and White Mountains, which form the eastern bounds of the Owens Valley and separate it from the Great Basin and Mojave Desert. The uplift of these mountains started between 2.8 and 2.3 mya (*Bachman, 1979*; *Lee et al., 2009*), and the calibration prior for the common ancestor of the three *Trogloderus* species distributed across these mountains was set as a normal distribution with a mean of 2.5 mya and standard deviation of one my. The second calibration is the deeply incised eastern margin of the Grand Canyon in northern Arizona. Two populations of a new species were sampled, one from sand dunes north of the Colorado river just below the Vermillion Cliffs, and one south of the Colorado River near Moenkopi. These two populations are separated by the gorge just downstream from Marble Canyon, which was been dated as 0.83 my old (*Polyak, Hill & Asmerom, 2008*). The calibration prior for the common ancestor of these two populations was set as a normal distribution with a mean of 0.83 mya and a standard deviation of 0.35 my.

Historical biogeographic reconstructions were performed in the BioGeoBEARS package (*Matzke, 2013*) in R (*R Core Team, 2018*) using the calibrated tree from the RelTime analysis with each species collapsed to a single tip. Six biogeographic regions were defined (Fig. 1) based primarily on previous biogeographic work of the intermountain (*Reveal, 1979*) and southwestern desert (*Van Dam & Matzke, 2016*; *Wilson & Pitts, 2010*) regions. The six areas are as follows: (1) Great Basin—centered around northern Nevada, northwestern Utah and southern Idaho in the regions shaped by the prehistoric lakes Lahontan and Bonneville and including the Snake River plain (*Reveal, 1979*; *Britten & Rust, 1996*; *Wilson & Pitts, 2010*); (2) Mojave Desert—the southwestern-most region of *Trogloderus* distribution which includes much of southeastern California, southern Nevada as well as far western Arizona and southwestern Utah (*Shreve, 1942*; *Reveal, 1979*; *Wilson & Pitts, 2010*; *Van Dam & Matzke, 2016*); (3) Lahontan Trough—a transverse transition zone between the Mojave and Great Basin deserts which shares floristic components with both regions and was never part of the prehistoric Lake Lahontan (*Reveal, 1979*; *Pavlik, 1989*; *Britten & Rust, 1996*; *Hafner, Reddington & Craig, 2006*); (4) Colorado Plateau—the desert areas surrounding the four-corners region west of the Rocky Mountains and generally east of the Wasatch mountains of Utah (*Reveal, 1979*; *Wilson & Pitts, 2010*); (5) Owens Valley—a narrow region bounded by the eastern Sierra Nevada mountains to the west and the Inyo and White mountains to the east, this transition region also has strong floral and faunal similarities with both the Mojave and Great Basin deserts (*Reveal, 1979*; *Andrews, Hardy & Giuliani, 1979*; *Macey, 1986*; *Pavlik, 1989*; *Van Dam & Matzke, 2016*); and (6) Widespread—this was used for outgroup taxa whose ranges do not coincide with the regions listed above and instead extend into other areas of western North America.

## Data management and availability

Label data from all specimens examined were digitized and are available online through the Symbiota Collections of Arthropods Network (SCAN; *Gries, Gilbert & Franz, 2014*, http://scan-bugs.org). Collecting events lacking GPS data on the label were georeferenced using Google Earth Pro version 7.3 and GEOLocate (www.geo-locate.org) as implemented in SCAN. Specimens from external institutions, which constituted the majority of those examined, were digitized using the SCAN Collection of Externally Processed Specimens (ARTSYS, see *Johnston, Aalbu & Franz, 2018*). All molecular and disarticulation vouchers are deposited in the MAJC and have images available with the pertinent specimen records on SCAN. Due to the fully digitized and available specimen data, verbatim label data are not included in the main text except for holotypes. Georeferenced specimen records were mapped using QGIS v3.2 which incorporated Google Earth satellite imagery.

Full locality, institutional ownership, determination and georeferencing data for all specimens studied are available as a csv file in Data S1. A Darwin-Core Archive of all digitized specimen data is available in Data S2. Full sequence alignments and configuration files for divergence analyses are available in Data S3. Individual sequences are also deposited on GenBank with accession numbers: MN597189–MN597394.

The electronic version of this article in Portable Document Format will represent a published work according to the International Commission on Zoological Nomenclature (*ICZN, 1999*, *2012*), and hence the new names contained in the electronic version are effectively published under that Code from the electronic edition alone. This published work and the nomenclatural acts it contains have been registered in ZooBank, the online registration system for the ICZN. The ZooBank Life Science Identifiers (LSIDs) can be resolved and the associated information viewed through any standard web browser by appending the LSID to the prefix http://zoobank.org/. The LSID for this publication is: urn:lsid:zoobank.org:pub:678EBFE3-6308-4FB8-8E93-184CEC9A15E7. The online version of this work is archived and available from the following digital repositories: PeerJ, PubMed Central and CLOCKSS.

## SYSTEMATICS

**Amphidorini LeConte, 1862**

The complex nomenclatural and taxonomic history of Amphidorini has been summarized by *Doyen & Lawrence (1979)* and *Johnston et al. (2015)*, and is only briefly described here. The tribe has frequently and historically been treated within the subfamily Tenebrioninae Latreille, 1802 (*Bouchard et al., 2005*, *2011*; *Bousquet et al., 2018*), but recent phylogenetic studies place the Amphidorini in a clade with several other tribes in what has been referred to as the subfamily Opatrinae Brullé, 1832 (*Aalbu et al., 2002*; *Kanda, 2017*; *Kamiński et al., 2018*).

The North American genera of this tribe can be separated from other members of Tenebrionidae by the following combination of characters: abdominal ventrites III–IV

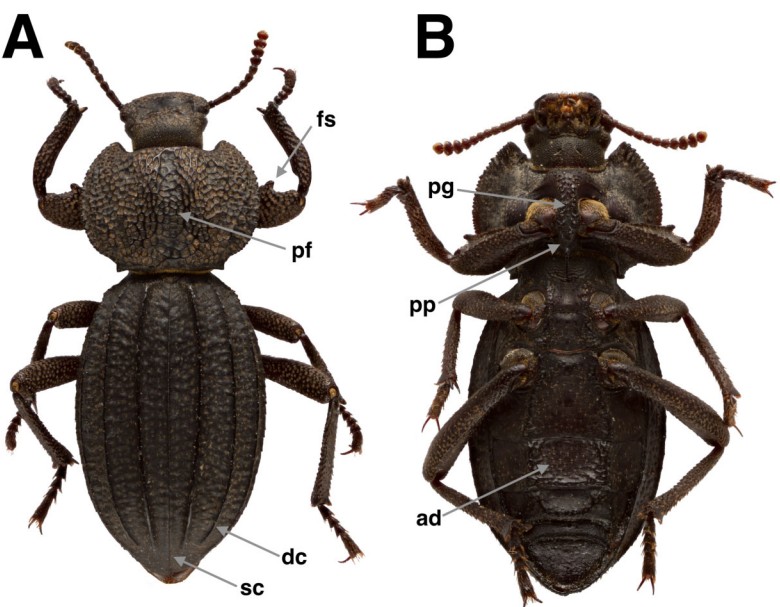

**Figure 2** *Trogloderus* **external morphology.** (A) Dorsal habitus, *Trogloderus vandykei* La Rivers. (B) Ventral habitus, *Trogloderus vandykei* La Rivers. ad, abdominal depression; dc, elytral discal costa; fs, femoral spine; pf, pronotal foveae; pg, prosternal groove; pp, prosternal process; sc, elytral sutural costa.

with visible membrane along hind margin; antennae lacking compound stellate sensoria; tarsal claws simple, not pectinate; penultimate tarsomeres not lobed beneath; elytra fused medially, hind wings reduced to small folds; paired defensive glands present between abdominal sternites VII and VIII, glands separate lacking a common volume, glands smooth, not annulated; mentum trilobed with mesal face more or less produced anterad, often concealing insertion of ligula; female paraproct and coxite short, coxite 1-segmented, with short subapical gonostyle; female with single, bursa-derived spermatheca.

The tribe is currently comprised of seven genera, six of which are known only from North America. Published keys to genera (*Aalbu et al., 2002*; *Johnston et al., 2015*) are sufficient to separate *Trogloderus* from other Amphidorini, though a generic revision of the tribe is in progress (MA Johnston & AD Smith, 2020, in preparation).

### *Trogloderus* LeConte, 1879

Type species *T. costatus* LeConte, 1879, by monotypy

**Diagnosis**. *Trogloderus* (Fig. 2) can be distinguished from other members of Amphidorini by the following characters: body roughly sculptured, pronotum either tuberculate or roughly punctured. Elytron with four sharply carinate longitudinal costae, elytral suture costate or not. Tarsi lined beneath with yellow to castaneus spicules, never with tomentose pads, probasitarsus thickened ventrally near distal margin.

**Male:** Body elongate, roughly sculptured, ferruginous to black. Length 9–16 mm. Width four to six mm.

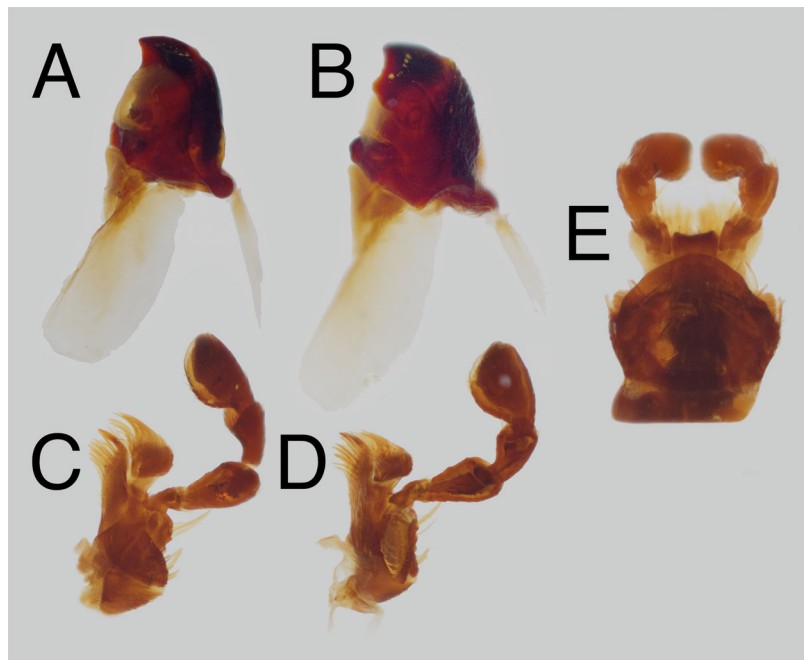

**Figure 3 *Trogloderus* mouthparts.** Dissected from MAJC0004230, *T. major* Johnston n.sp. (A) Left mandible, ventral view. (B) Right mandible, dorsal view. (C) Right maxilla, ventral view. (D) Left maxilla, dorsal view. (E) Labium, ventral view.

Head. As broad as long. Antenna 11-segmented, extending to posterior 2/3 of pronotum; antennomere III 1.5× as long as IV, IV–VII obconical, roughly as long as wide, VIII–XI wider than long, VIII with sensory patch of yellow setae along outer margin of apical face, IX–XI with sensory patch forming continuous ring around apical face. Labrum free, partially exposed, broader than long; anterior margin rounded laterally, deeply sinuate mesally; each lobe bearing tuft of short yellowish setae; dorsal surface punctate, each puncture bearing a long yellow seta, punctures becoming denser anteriorly; hypopharynx originating just posteriorly of anterior ventral margin, anterior hypopharyngeal sclerite ovoid, transverse, 1.5× wide as long. Mandibles (Figs. 3A and 3B) roughly symmetrical, bidentate, the left slightly larger than and overlapping the right at rest; dorsal face striate, more strongly so anterolaterally; lateral face with longitudinally elongate punctures, each bearing a single seta; ventral surface concave, smooth; mola present, strongly sclerotized, finely granulate; prostheca large, membranous, extends laterally around mola to form a large submola. Maxillae (Figs. 3C and 3D) ferruginous, symmetrical; cardo subtriangular, narrowing proximaly, lightly punctate; basistipes bearing thickened setae, subtriangular, narrowing distally, articulated with cardo basally and basigalea anterolaterally, mediostipes anteromesolly, and palpifer anterolaterally; mediostipes subtransverse, glabrous, articulated with lacinia distally; lacinia well developed, mesal surface bearing a terminal digitus followed proximally by robust lacinial teeth which become setae in basal 1/3; basigalea thin, articulated with distigalea apically, bearing fine, long setae; distigalea 1.5× longer than wide, anterior and mesal surface densely clothed with thick yellowish setae, dorsal surface bearing moderately separated,

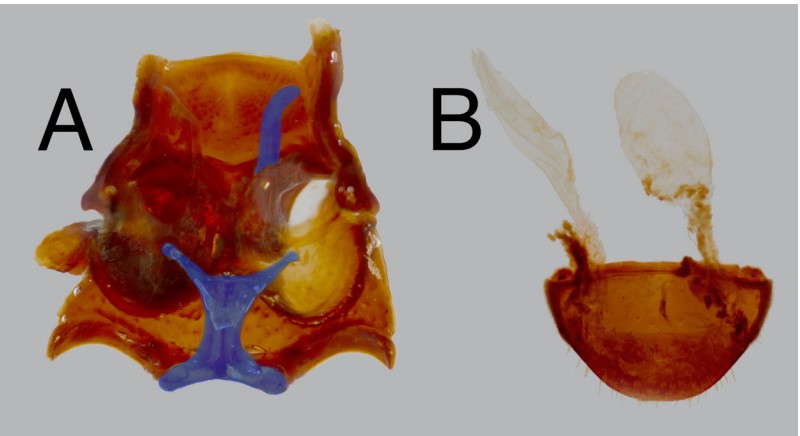

**Figure 4 *Trogloderus* internal morphology.** (A) Pterothorax venter, dorsal internal view; Metendosternite and right mesosternal apophysis highlighted; Dissected from MAJC0004244, *T. warneri* Johnston n.sp. (B) Defensive glands and abdominal ventrite V, dorsal internal view; Dissected from MAJC0004231, *T. arcanus* Johnston n.sp.               

long yellowish setae; palpifer digitate ventrally, bearing stout setae; palpi with four palpomeres, palpomere I small, subtriangular, II elongate, obconical, III slightly shorter than II, clavate, IV securiform, apical surface bearing yellowish membranous sensorium. Mentum (Fig. 3E) trilobed, with mesal region of dorsal face produced anteriorly into arcuate lobe, covering insertion of ligula; ligula transverse, bearing two apical tufts of stout setae along dorsal face; labial palp with three palpomeres, palpomere I obconical, as long as wide, II clavate, 1.5× long as wide, III fusiform and evenly setose; hypopharyx moderately sclerotized along anterior margin, hypopharyngeal brush forming thickened longitudinal band from anterior margin of hypopharynx to posterior margin of mentum. Clypeus fused to frons, roughly sculptured, broadly sinute at middle, frontoclypeal suture indistinct to faintly traceable in teneral individuals. Frons usually slightly sunken, less roughly sculptured than clypeus, with slightly elevated bilobed tubercle centrally; epistomal lobes produced, distinctly offset from clypeus. Eyes entire, reniform, dorsal lobe 5–6 facets wide, ventral lobe 3 facets wide. Vertex at same level and contiguous with central tubercle of frons; becoming strongly granulate toward occiput. Submentum short, arcuate posteriorly, faintly evident; gular sutures diverging posteriorly, well rounded, gula less coarsely sculptured than surrounding head capsule.

Thorax. Pronotum roughly sculptured; lateral margins strongly curved, crenulate along entire length, sinuate at posterior angle, anterior angles acute, projected, with longitudinal depression along midline, often separated into anterior and posterior foveae (Fig. 2A, pf); prosternal length from anterior margin to procoxae subequal to procoxal diameter; procoxae separated by approximately ½ procoxal diameter; prosternal process (Fig. 2B, pp) projected posteriorly; procoxal cavities closed posteriorly by postcoxal bridge of pronotum which meets the prosternal process mesally; pleural apophysis (Fig. 4A) directed anterodorsally, becoming laminar and longitudinally expanded near ventral surface of pronotum, with short dorsal coxal articulation extended mesally around basal 1/3; prosternal apophysis straight, extending dorsolaterally, terminating near dorsal margin of

coxa. Mesonotum strongly transverse, triangular, densely papillose; scutellar shield wide and short, lacks papillae, strongly microsculptured; mesanepisternum subtriangular, narrowing posteroventrally, anterior 1/3 with integument thickened, papillose, offset from posterior 2/3 by posteriorly concave ridge demarking a section of thinner integument, punctate in posterior 1/3; mesepimeron short, fairly evenly punctate; mesoventrite with anterior 1/2 covered by prothorax at rest, posterior 1/2 projected ventrally between coxae, with longitudinal groove to receive prosternal process; mesocoxal cavities closed externally by mesoventrite, mesepimeron, and metaventrite; mesosternal apophyses extend anteriorly from apex of mesocoxal cavity, recurved dorsally and then posteriorly around anterior 1/4 of mesoventrite. Metanotum greatly reduced, prescutum forming narrow arch, strongly connected to the mesonotum, remainder of metanotum forming short, somewhat heavily sclerotized membrane, without discernable subregions; metepimeron forming narrow rod-like longitudinal sclerite along length of metathorax, concealed beneath elytron, posteriorly with short ventrally projected metepimeral process which is fused with metepisternum above metacoxal cavity; metepisternum elongate, subrectangular; metaventrite short, length less than mesocoxal diameter, antecoxal ridge deeply impressed above anterior coxal margin, discrimen not apparent; metacoxal cavities closed externally by metaventrite, metepisternum, metepimeral process, and first abdominal ventrite; metendosternite (Fig. 4A) stout, stalk broad, ventral longitudinal flange very well sclerotized, furcae as wide as stalk, relatively immovable, furcal apicies reflexed posterolaterally, forming horizontal pad for furca-trochanteralis muscle attachment, anterior tendons inserted at apical 1/4 of furcae. Elytra fused, suture elevated or not; elytron disc (Fig. 2A, dc) with four longitudinal carinate costae; epipleuron narrow throughout length, not or slightly widened anteriorly, attaining elytral apex posteriorly. Hind wings greatly reduced, forming veinless tubular sac, approximately the size of first abdominal spiracle.

Legs. Fore leg slightly enlarged, weakly fossorial; femur clavate, heavily punctate, dorsal anterior margin carinate from base to apical 1/5, ending in short recurved spine (Fig. 2A, fs); tibia with inner face excavated in basal 1/5, outer face carinate from base to near tarsal insertion, apex bearing row of ferruginous spicules dorsally, tibial spurs subequal, extending to apex of tarsomere II; tarsus bearing furriginous spicules, tarsomere I ventrally thickened at apex, maximum height equal to length, II–IV subequal, relatively short, about as tall as long, V slightly clavate, as long as II–IV combined; empodium minute, hidden within tarsal apex, bearing two yellowish setae; tarsal claws simple, evenly arcuate, 2/3 length of tarsomere V. Middle and hind legs similar to fore leg, tibia subcylindrical, not expanded; all tarsomeres simple, not thickened beneath.

Abdomen. Five visible ventrites, ventrite I intercoxal process truncate, rectangular, twice as broad as long, I–III connate, fused to elytra laterally, III–IV with visible membrane posteriorly, I–II bearing variously developed longitudinal ridges demarking flattened abdominal depression (Fig. 2B, ad) in line with thoracic intercoxal region; tergites membranous, weakly sclerotized; paired defensive glands (Fig. 4B) present posterior to ventrite V (between sternites VII–VIII), glands lacking a common volume, each gland

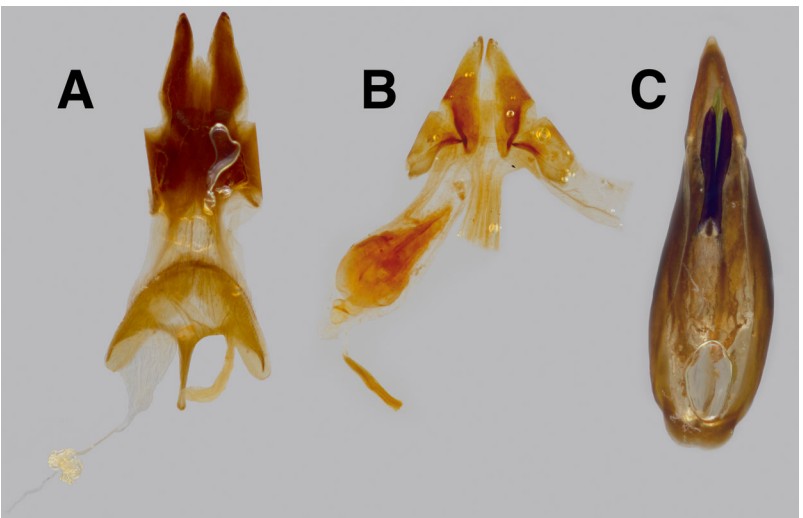

**Figure 5 *Trogloderus* terminalia.** (A) Female terminalia, dorsal view, showing bursa-derived spermatheca; *T. vandykei* La Rivers. (B) Female terminalia, ventral view, showing bursa copulatrix and oviduct; Dissected from MAJC0004243, *T. major* Johnston n.sp. (C) Male adeagus, ventral view; clavae and penis highlighted; *T. vandykei* La Rivers.                     

elongate, subfusiforme, extending anterior of ventrite II, membrane finely strigose, lacking annular pleats, gland openings centered around lateral 1/5.

Terminalia. Tergite VIII weakly sclerotized, posterior margin evenly arcuate, bearing row of fine golden setae; sternite VIII weakly sclerotized, bilobed, deeply emarginate posteriorly, each lobe subtriangular, clothed ventrally and posteriorly with long yellowish setae, anterior deeply margin bisinuate, thickened into apodemes. Spicules V-shaped, fused anteriorly, 1.5× length of tergite VIII, spicule plates moderately small, 4× width of spicules, twice as long as wide. Adeagus (Fig. 5C) elongate, cylindrical; basal piece 4× as long as wide, lateral margins (alae) reflexed inwardly, leaving ventral face open, apicodorsal margin concave; parameres fused, 1/2 length of basal piece, widest basally, 1.5× long as wide, apical half curved ventrally; clavae (Fig. 5C) narrow, about as long as parameres, 1/6 maximum width of parameres; penis narrow, lightly sclerotized, fully hidden dorsally by parameres at rest.

**Female.** As male but generally stouter, fore femoral spines variable, typically less developed than those of males, base of tibia generally not constricted, central abdominal groove less developed.

Terminalia. Tergite VIII moderately sclerotized, posterior margin evenly arcuate, bearing golden setae; sternite VIII moderately sclerotized, evenly arcuate podsteriorly, bearing golden setae, fused medially to spiculum ventrale along anterior margin, spiculum ventrale 1.5× medial length of tergite VIII. Proctiger (Fig. 5A) slightly longer than wide, posterior margin weakly emarginate, bearing single row of short yellow setae. Paraproct subrectangular dorsally (Fig. 5A), subtriangular ventrally (Fig. 5B), bacculus obliquely pointed psoteromesally, thickened mesally. Coxite 1-segmented, subrectangular in dorsal view, narrowing posteriorly, subtriangular in ventral view, bacculus obliquely pointed

anteromesally. Gonostyle short, inserted ventrally, at most weakly visible from above. Bursa copulatrix (Fig. 5B) about 2× length of coxite, bearing single spermatheca (Fig. 5A) off of duct from anterior margin with single long spermathecal gland.

## Variation and natural history

Sexual dimorphism is primarily observed in the fore tibiae and abdominal ventrites. The fore tibiae of males are generally more explanate along the outer edge and are more strongly constricted proximally. The femoral spines are often slightly stronger in the males as well, where they pair with the constricted tibiae to form a grasping mechanism—presumably used to hold the females legs or antennae during copulation. The abdominal depression also tends to be stronger in males, with the marginal ridge more produced and the central region more depressed. This is also assumed to help the male in positioning during copulation.

Relatively little is known of *Trogloderus* biology. Adults have not been successfully cultured in the lab and larvae and pupae remain unknown and undescribed from the wild. Adult beetles are able to burrow into loose sand, where the immature stages presumably live. More commonly, adults are observed emerging from mammal burrows after dark where they seem to take shelter underground during the day. Like other Amphidorini, adults can also be found, though not particularly abundantly, under rocks or loose boards. *Trogloderus* are very active at night, and seem to travel good distances across open ground likely in search of food, mates, or new sites to shelter during the day.

Collection records and field observations indicate that this genus is restricted to habitats with loose sand. While the largest populations seem to be from deep aeolian sand formations, they can also be found in areas of fine loose sand along rivers and across desert flats, for example, in small sand hummocks around the base of desert shrubs.

## Key to the species of *Trogloderus*
### Diagnostic utility of characters

The extreme sculpturing of *Trogloderus* makes the genus readily recognizable among Amphidorini, but also seems to magnify—in the context of species identification—the relatively broad individual and geographic intraspecific variation found throughout the tribe (*Triplehorn & Thomas, 2012*; *Johnston, 2015*, *2016*). The female ovipositor has been heavily relied upon to classify species into genera and subgenera (*Blaisdell, 1909*; *Triplehorn & Thomas, 2012*; *Johnston, 2015*, *2016*), yet it is fairly constant throughout *Trogloderus* and was found unreliable for species identification. Male terminalia can be diagnostic for some species, but not for all (*Somerby, 1972*; *Aalbu, Smith & Triplehorn, 2012*). Within *Trogloderus*, the basic shape of the parameres can sometimes aid in distinguishing some species from each other by examining the curvature of the lateral margins, but do not alone reliably distinguish one species from all others.

General facies, elytral sculpturing, and body size were found to be largely unreliable for species recognition as they can vary within populations and especially between populations. It is not uncommon to find locally homogenous populations to have strong differences between them. Whether this is due to some environmental variable such as

food or water availability or simply stochastic due to limited gene flow is unclear. The sculpturing of the pronotum and head seems to be more stable within species and are heavily relied upon in the following identification key.

Though coloration was previously used as a secondary diagnostic character (*La Rivers, 1946*; *Papp, 1961*), it is here found to be unusable for species determinations. Rather, it seems that the cuticle of adult *Trogloderus* takes a fairly long time to fully harden and that more teneral specimens exhibit a red coloration, which then matures to a darker black in the longest-lived individuals. This is based on the observation that in almost every large series known there is a spectrum of red to castaneus to black individuals. Specimens with a brighter red coloration seem to have thinner cuticle (personal observation while pinning specimens) and even less strongly sclerotized terminalia. This is perhaps a strategy for these desert-dwelling beetles to limit the duration of the potentially more susceptible immature stages in preference of a longer hardening period as an adult. It is not clear whether the teneral adults are reproductively viable as no eggs have been observed in such individuals when dissected, and this could be an example of Reifungsfraß, the need for a maturation feeding period (see *McNee, Wood & Storer, 2000*).

Dichotomous key to the species of adult *Trogloderus*

1 Pronotal surface distinctly tuberculate.............................................. 2

1′ Pronotal surface not tuberculate, heavily punctate to cribrate ....................... 5

2 (1) Each elytron with large subapical tubercle at outer carinal terminus; pronotal foveae delimited laterally by raised longitudinal ridges (Mojave Desert).......................
............................................ *Trogloderus tuberculatus* Blaisdell (Fig. 6A)

2′ Elytra without posterior tubercles; pronotum lacking elevated ridges, foveae lined by tubercles originating from same surface as those of the disc (widespread) .............. 3

3 (2′) Posterior pronotal angles more or less inflated; lateral margins of pronotal disc slightly depressed, lacking tubercles (western Colorado Plateau) .........................
...................................... *Trogloderus warneri* n. sp. (Fig. 6C)

3′ Posterior pronotal angles not at all inflated; lateral regions of pronotal disc not depressed, tubercles relatively evenly dispersed from foveae to lateral margins ......... 4

4 (3′) Male parameres triangular, evenly tapering from base to apex; elytral carinae often granulately tuberculate on sides (west of Kaibab Plateau) ...............................
...................................... *Trogloderus skillmani* n. sp. (Fig. 6D)

4′ Male parameres distinctly constricted near base, then evenly tapering to apex; elytral carinae usually lacking tubercles on sides (east of Kaibab Plateau)*Trogloderus verpus* n. sp. (Fig. 6B)

5 (1′) Pronotal dorsum bilobed in anterior view; pronotum strongly explanate laterally; pronotal foveae joined into single longitudinal groove (Mojave Desert)..................
...................................*Trogloderus vandykei* La Rivers (Fig. 1)

5′ Pronotum evenly convex in anterior view; pronotum weakly to moderately explanate laterally; pronotal foveae variable, often distinctly separated (widespread) .............. 6

6 (5′) Pronotum cribrately punctured, margins of punctures strongly elevated; intervals between elytral carinae bearing short, transverse secondary ridges......................
......................................... *Trogloderus costatus* LeConte (Fig. 6F)

6′ Pronotum heavily punctate, margins of punctures not strongly elevated; intervals between elytral carinae usually smooth, lacking well-defined secondary ridges.......... 7

7 (6′) Propleurae lacking tubercles on dorsal half, never with tubercles anteriorly just underneath pronotal margin; pronotal foveae joined into single well-demarked longitudinal groove ................................... *Trogloderus major* n. sp. (Fig. 6H)

7′ Propleurae wth tubercles in dorsal half, at least anteriorly underneath pronotal margin; pronotal foveae variable, usually not forming single longitudinal groove............... 8

8 (7′) Epistoma roughly punctured, individual punctures evident above antennal insertion; pronotal punctures fairly evenly circular, discrete; elytral costae moderately to strongly produced; male parameres broadly triangular in dorsal view, sides straight and evenly tapered (southern Owens Valley)..................... *Trogloderus kandai* n. sp. (Fig. 6I)

8′ Epistoma finely to roughly tuberculate, individual punctures not evident above antennal insertions; pronotal punctures often longitudinally oval, sometimes coalescent anteriorly; elytral costae weakly to moderately produced; male parameres narrowly triangular in dorsal view, sides gently to moderately arcuately concave (widespread)................. 9

9 (8′) Frontoclypeal suture forming a complete transverse ridge, frons apex below the plane of clypeus base; male parameres broadly triangular, evenly converging; prosternal process horizontal, on the same plane as the prosternum between the procoxae; punctures larger (northern Great Basin) ........................ *Trogloderus nevadus* La Rivers (Fig. 6G)

9′ Frontoclypeal suture usually not forming complete transverse ridge, mesal region of frons apex on the same plane as clypeus; male parameres usually noticeably constricted near base, with sides slightly convexly arcuate; prosternal process often narrowed at posterior procoxal margin, sometimes dorsally offset from plane of prosternum; pronotal punctures usually smaller (Lahontan Trough including Mono Lake region of Owens Valley) ........................................... *Trogloderus arcanus* n. sp. (Fig. 6E)

### *Trogloderus arcanus* Johnston, New Species

urn:lsid:zoobank.org:act:0BCDA9E8-F615-41B9-9376-62778B0958EE

Figures 6E, 7A and 8

**Diagnosis**. *Trogloderus arcanus* can be distinguished from all congeners, except *T. nevadus*, by the combination of tuberculate propleurae, frons, and clypeus. To separate it from the latter, the characters given in the key will usually separate the two species, but see the variation and remarks below.

**Description.** As genus with the following: Length 7.0–10.5 mm, width 3.5–4.5 mm. Head. Epistoma and frons tuberculate, lacking distinct punctures; mesal region of frons elevated, usually on same plane as clypeus, rendering transverse ridge along frontoclypeal suture incomplete, lateral regions of frons usually evenly tuberculate. Thorax. Pronotum

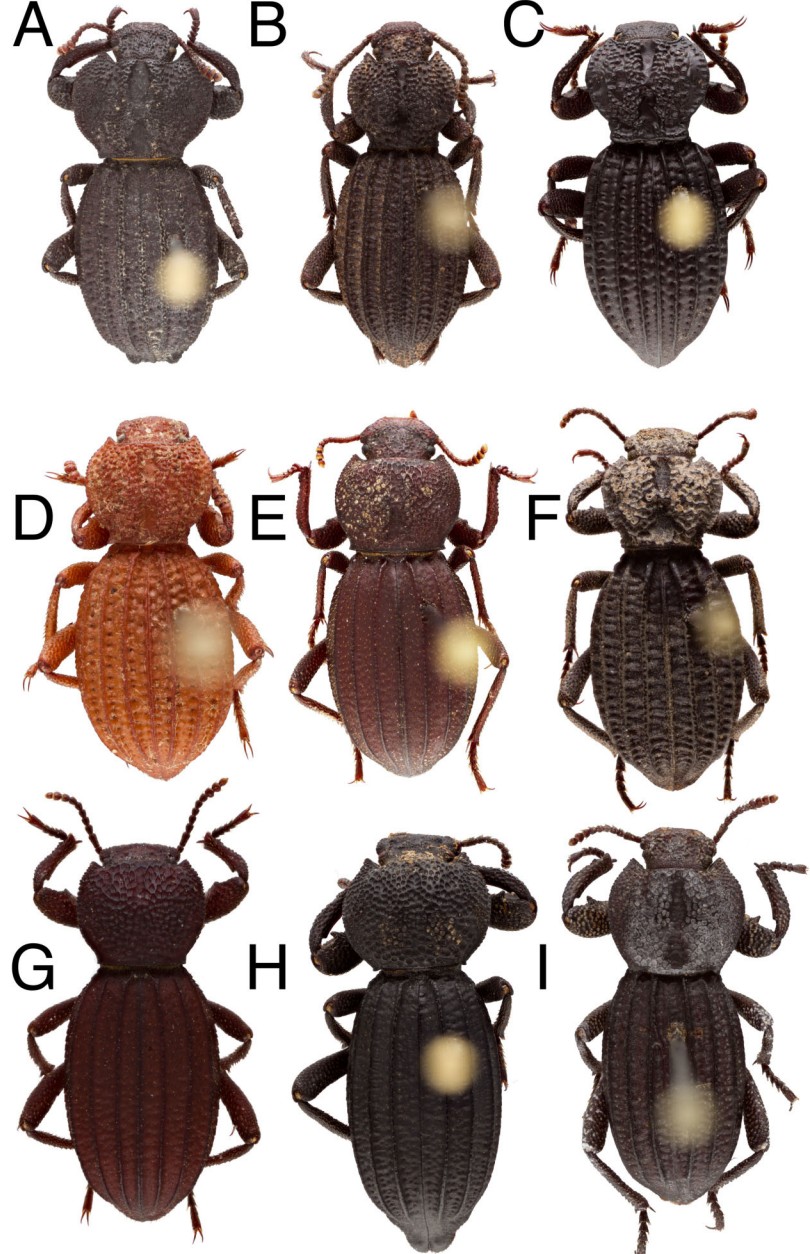

**Figure 6 _Trogloderus_ species, dorsal habitus.** (A) _T. tuberculatus_ Blaisdell (non-type). (B) _T. verpus_ Johnston n.sp. (holotype). (C) _T. warneri_ Johnston n.sp. (holotype). (D) _T. skillmani_ Johnston n.sp. (holotype). (E) _T. arcanus_ Johnston n.sp. (holotype). (F) _T. costatus_ LeConte (non-type). (G) _T. nevadus_ La Rivers (non-type). (H) _T. major_ Johnston n.sp. (holotype). (I) _T. kandai_ Johnston n.sp. (holotype).

evenly convex dorsally; heavily punctate, punctures longitudinally elongate, tending to coalesce anteriorly; lateral margins moderately arcuate, sinuate along basal fifth; posterior pronotal angles obliquely acute, relatively small; anterior fovea usually obsolete to moderately impressed, posterior fovea always distinct, round, deeper than anterior. Propleurae usually tuberculate throughout, tubercles always present anteriorly underneath

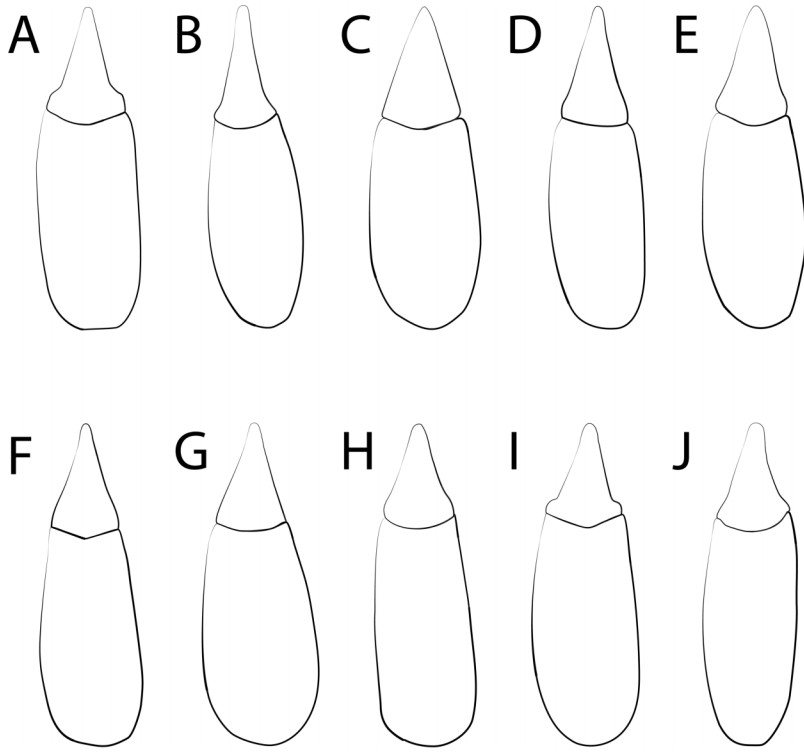

**Figure 7** *Trogloderus* **species, adeagus dorsal view.** (A) *T. arcanus* Johnston n.sp. (B) *T. costatus* LeConte. (C) *T. kandai* Johnston n.sp. (D) *T. major* Johnston n.sp. (E) *T. nevadus* La Rivers. (F) *T. skillmani* Johnston n.sp. (G) *T. tuberculatus* Blaisdell. (H) *T. vandykei* La Rivers. (I) *T. verpus* Johnston n.sp. (J) *T. warneri* Johnston n.sp.  

pronotal margin. Prosternal process usually narrowed along posterior procoxal margin, often narrowed and on slightly dorsal plane than prosternum between procoxae. Elytral costae weakly to moderately developed, intervals usually smooth, occasionally with slight transverse ridges; elytral suture elevated along poster half, nearly as prominent as discal costae. Abdomen. Abdominal depression relatively weak, usually not discernable on ventrite II. Male terminalia. Parameres (Fig. 7A) usually appearing narrow, arcuately constricted near base, sides usually slightly concave, occasionally appearing roughly evenly triangular.

**Variation.** The diagnostic characters of this species are quite variable both within and between populations. Specimens from Crescent Dunes south to Sarcobatus Flats tend to have a distinctly narrowed prosternal process, while specimens from Teel's Marsh and Silver Peak west to Mono Lake tend to have a horizontal, evenly narrowing prosternal process. Specimens from lower elevation regions (typically Nevada) are fairly weakly sculptured, having rather small pronotal punctures, sometimes becoming separated by as much as half of their diameter, and fairly weakly developed elytral costae. Specimens from higher elevation (e.g., Mono County, CA) tend to be more roughly sculptured on the pronotum and elytra. The latter populations tend to also have the frontoclypeal ridge more or less complete throughout. The northern and eastern populations (e.g., Crescent Dunes

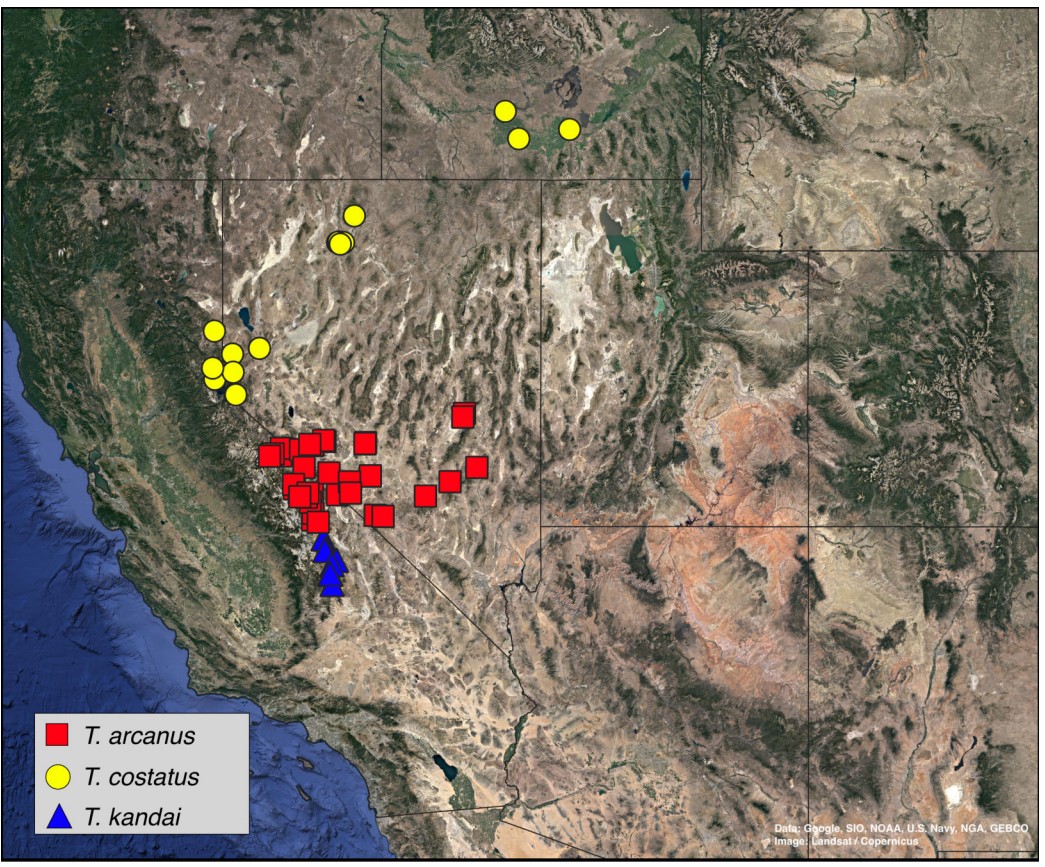

**Figure 8 Distribution map, *Trogloderus arcanus* Johnston n.sp., *T. costatus* LeConte, *T. kandai* Johnston n.sp.** Map data ©2019 Google, SIO, NOAA, U.S. Navy, NGA, GEBCO.

and Coal Valley) have more distinctly narrowed parameres, while the southern populations (e.g., Silver Peak) tend to have slightly broader and more evenly tapered parameres.

**Distribution.** This species is distributed throughout the region known as the Lahontan Trough (*Reveal, 1979*), a region which was never part of the prehistoric Lake Lahontan to the north (Fig. 8).

**Type material.** Holotype. "USA: NEV: Nye Co., 12 mi/NW Tonopah, Crescent/Dunes; 38°13′47″N, 117°/20′06″W; JUN 30-JUL 9/2011; barrier pitfalls w./fish bait; W.B. Warner," "ARTSYS0007057" bearing red holotype label. Deposited in ASUHIC, catalog number ASUHIC0101562. Paratypes. A total of 765 specimens from throughout the range bearing blue paratype labels (see Data S1 and S2 or SCAN for full specimen data).

**Etymology.** The specific epithet, meaning secret, or mysterious (*Brown, 1956*), is given for this cryptic species that was very difficult to separate from *T. nevadus* and was first revealed as a distinct species through the molecular phylogeny presented below.

**Remarks.** The geographically linked morphological variation in this species warrants further study, which will rely on increased collecting efforts in an under-collected region and likely more molecular data. The slightly heterotypic species as circumscribed here may represent a cryptic species complex. Strong differences between populations may be the result of reproductive isolation and diverging evolutionary lineages, or could be linked to environmental conditions. The roughly sculptured populations from California are from cooler and more mesic habitats, whereas the central Nevada populations face much drier and warmer conditions. There also may be some competitive exclusion or prezygotic isolation pressures, which shape the Nevada populations that border along the range of *T. nevadus.*

### *Trogloderus costatus* LeConte, 1879

Figures 6F, 7B and 8

**Diagnosis.** *Trogloderus costatus* can be easily separated from all congeners by the cribrately punctate pronotum, where the margins of the punctures are strongly elevated. The presence of transverse ridges in the intervals of the elytral costae can also separate this species from any others with punctate pronota.

**Redescription.** Length 10.5–12 mm, width 4–4.75 mm. Head: Epistoma and frons roughly punctured to tuberculate; frontal tubercle usually roughly punctured, punctures usually becoming discrete tubercles toward clypeus; frontoclypeal suture forming gentle, complete transverse ridge. Thorax: Pronotum relatively evenly convex dorsally; cribrately punctured, punctured region elevated above less punctate lateral margins; anterior and posterior foveae very distinct, deep, impunctate; lateral margins evenly arcuate, recurved just before posterior angles; posterior angles obliquely acute, small. Propleurae distinctly and evenly tuberculate throughout. Prosternal process horizontal, forming short, evenly tapered triangle behind posterior procoxal margin. Elytral costae strongly developed, intervals always with distinct transverse ridges; elytral suture strongly elevated, nearly as prominent as discal coxae along posterior 5/6. Abdominal depression weak, not evident in females, occasionally evident on anterior 1/2 of ventrite I in males. Male terminalia. Parameres (Fig. 7B) narrow, arcuately constricted near base, sides concave, weakly arcuately converging to apex.

**Variation.** As with most other species, the intensity of the body sculpturing is variable both between and within populations. Specimens from near Winnemucca tend to have the weakest sculpturing, though the strongly elevated punctate regions of the pronotum are still diagnostic. *Trogloderus costatus* has the most variable cephalic sculpturing within the genus, with specimens ranging from having the entire dorsal aspect of the head distinctly punctate (Truckee river near Reno) to specimens that possess nearly entirely tuberculate heads (Winnemucca). Specimens from other regions have a mixture of both, generally with the frontal tubercle punctate and the punctures becoming distinct tubercles toward the clypeus.

**Types.** Holotype male from Rock Creek Owyhee County, Idaho at the Museum of Comparative Zoology, type number 4624, pictures available on-line from MCZ type specimen database. *LeConte (1879*: 3*)* specifically references "one specimen kindly given me by Mr. Reinecke; others are in the collections of Dr. Horn and Mr. Bolter." This statement is here interpreted to comply with the *ICZN (1999)* Article 73.1.1 and the above single specimen is considered the holotype upon which the nominal species was founded, with the secondarily mentioned specimens considered as paratypes.

**Material examined.** 63 specimens (see Data S1 and S2 or SCAN for full specimen data).

**Distribution.** This species is known from the Northern Great Basin, from regions once dominated by the prehistoric Lake Lahontan through the Snake River Plain (Fig. 8).

**Remarks.** This is the second least abundant species found in natural history collections, yet was the first species described in the genus. While true *Trogloderus costatus*, as recircumscribed here, is uncommon in collections, most existing specimens are determined to this species likely following the treatment of *La Rivers (1946)*. *T. costatus* overlaps most of the range of *T. nevadus*, though the latter is much more frequently collected. Specimens of *T. costatus* seem to retain the most substrate on their cuticle among all of its congeners, and perhaps this cryptic lifestyle makes it less commonly collected, or perhaps this morphological sculpturing is adapted to more specific substrates. With relatively few specimens known, and many of them lacking very precise locality data, increased collecting efforts may help elucidate drivers of this species' distribution and intense morphological sculpturing.

### *Trogloderus kandai* Johnston, New Species
urn:lsid:zoobank.org:act:09FCBD7E-3DE8-40E4-8C13-5EC6DFA734EA

Figures 6I, 7C and 8

**Diagnosis.** *Trogloderus kandai* can be separated from its congeners by having the pronotum punctate, propleurae tuberculate, and the epistoma distinctly punctured, at least above the antennal insertions. Most similar to *T. arcanus*, particularly specimens from the Mono Lake region, *T. kandai* can be further separated from the latter by the pronotal punctures being nearly evenly round and not tending to coalesce (longitudinally oval and tending to coalesce anteriorly in *T. arcanus*).

**Description.** Length 9–11 mm, width 3.5–4.5 mm. Head: Epistoma aspirately punctate, distinctly so above antennal insertions, often becoming somewhat tuberculate mesally; frontoclypeal suture forming complete transverse ridge; frons irregularly tuberculate, frontal tubercle fairly distinctly punctate, lobes connected by anterior transverse ridge. Thorax: Evenly convex dorsally; heavily and evenly punctate throughout, punctures round, not becoming coalescent, occasionally slightly elongate near anterior margin; lateral margins evenly arcuate, recurved just before posterior angles; posterior angles obliquely acute, small; anterior fovea usually forming moderately and evenly impressed longitudinal channel connected to posterior fovea, posterior fovea round, deeper than anterior fovea.

Propleurae tuberculate, tubercles often obscure posteriorly, always with tubercles anteriorly underneath pronotal margin. Prosternal process horizontal, forming evenly tapered triangle behind posterior procoxal margin. Elytral costae moderately produced, intervals punctate but lacking well developed transverse ridges; elytral suture usually not elevated basally, somewhat elevated in posterior 1/2 but less produced than discal costae. Abdomen: Abdominal depression moderately developed in both sexes, distinctly present on ventrites I–II, smoother than lateral region of ventrite in males, entire ventrite fairly similarly sculptured in females. Male Terminalia. Parameres (Fig. 7C) somewhat broad, evenly tapering from base to apex.

**Variation.** This species exhibits relatively constant morphology, perhaps due to the extremely limited known distribution. The sculpturing of the epistoma can be fairly variable within the population, but individual punctures can be observed along the outer edge above the antennal insertion. The elytral suture is also somewhat variable, usually being elevated in the posterior half, it is occasionally elevated along most of its length.

**Distribution.** This is the most geographically restricted species of *Trogloderus* and is known only from the southern Owens Valley in California, in the region around Owens Lake between independence and Olancha (Fig. 8).

**Type material.** Holotype. "USA:CA:Inyo Co./Olancha Dunes OHV area/N36°17.665′ W117°59.191′/3,600 ft. KK07_028/K. Kanda, 22.vii.2007," "ARTSYS0007058" bearing red holotype label. Deposited in the ASUHIC, catalog number ASUHIC0101561 Paratypes. A total of 82 specimens bearing blue paratype labels (see Data S1 and S2 or SCAN for full specimen data).

**Etymology.** I am pleased to name this species after the tenebrionid specialist Kojun Kanda, who both collected the holotype and provided direction on the molecular analyses.

**Remarks.** The restricted distribution of this species is very interesting, being bounded by the Coso Range to the south and a series of old lava flows to the north which are part of the southern boundary for the Tinemaha Reservoir. South of the Coso Range is traditional Mojave Desert habitat and is dominated by creosote bush (*Larrea tridentata* (DC.) Coville) which is only sporadically present to the north, largely replaced by the Great basin indicative big sagebrush (*Artemisia tridentata* Nutt.). Thus, *T. kandai* is only known from a transition region between the Mojave and Great Basin deserts.

### *Trogloderus major* Johnston, New Species

urn:lsid:zoobank.org:act:1B61B89E-5839-47CA-AA20-F4795FF931D7

Figures 6H, 7D and 9

**Diagnosis.** This species can be recognized by having a punctate and evenly convex pronotum, and the propleurae lacking tubercles on the dorsal half (if propleural tubercles present, they are located on the bulging region covering the procoxae). This species can be further separated from most other species with punctate pronota by the smooth elytral

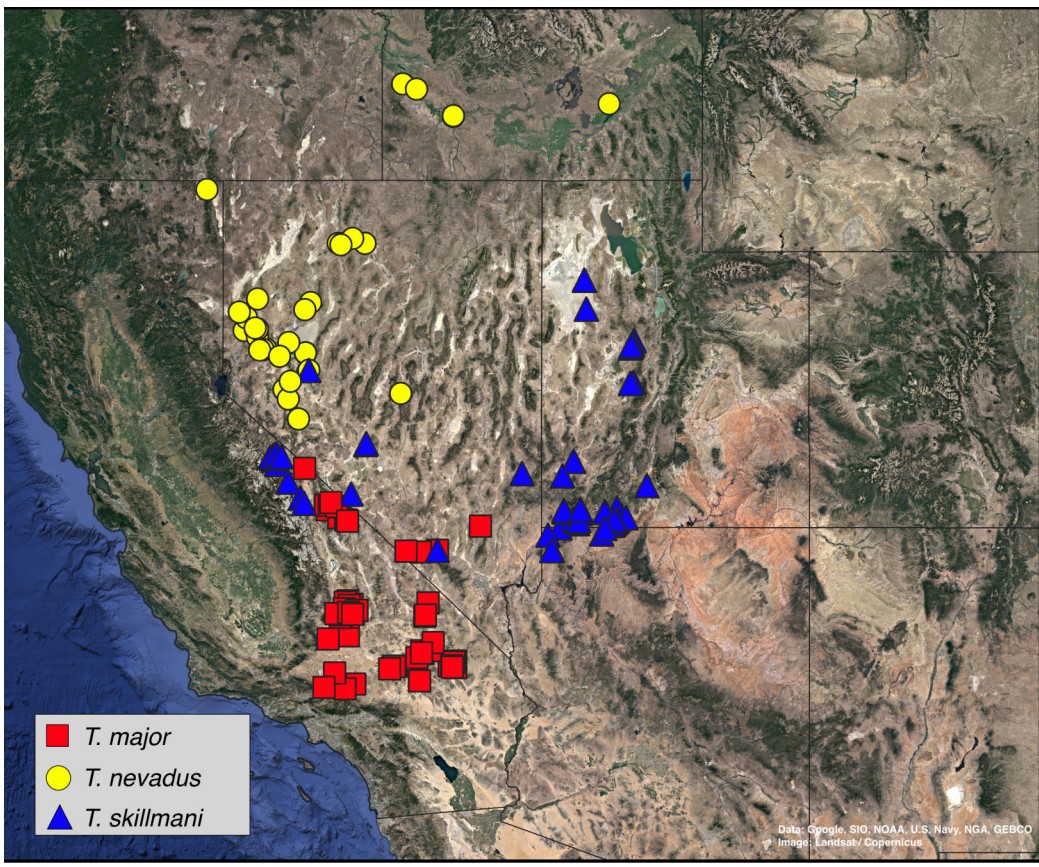

**Figure 9 Distribution map, *Trogloderus major* Johnston n.sp., *T. nevadus* La Rivers, *T. skillmani* Johnston n.sp.** Map data ©2019 Google, SIO, NOAA, U.S. Navy, NGA, GEBCO.

suture, located in a depressed interval between the inner elytral costae. This form of the elytral suture and propleurae lacking tubercles is shared with the sympatric species *T. vandykei*, which has a bilobed dorsum of the pronotum in anterior view.

**Description.** As genus with the following: Length 9.5–13.5 mm, width 4–5.5 mm. Head: Epistoma usually distinctly punctured, sometimes becoming irregularly tuberculate mesally; frontoclypeal suture forming complete transverse ridge; frontal tubercle punctate, lateral regions of frons smooth. Thorax: Pronotum evenly convex dorsally; heavily and evenly punctate throughout; lateral margins fairly evenly arcuate, recurved just before posterior angles; posterior angles obliquely acute, very small; anterior fovea forming weakly to moderately impressed longitudinal channel, connecting to posterior fovea; posterior fovea round, moderately impressed, slightly deeper than anterior fovea. Propleurae lacking punctures on dorsal half, always lacking punctures anteriorly underneath pronotal margin, usually with indistinct tubercles on inflated region covering procoxal cavity. Prosternal process robust, horizontal, forming evenly tapered triangle behind procoxal posterior margin. Elytral costae weakly to moderately elevated, intervals relatively smooth, bearing faint traces of transverse ridges; elytral usually suture not

elevated, or if elevated posteriorly then significantly shorter than the discal costae. Abdomen: Ventrites relatively smooth laterally; abdominal depression strong, distinct in both sexes, stronger in males, margins of depression roughly punctured, depression distinctly margined throughout ventrites I–II; ventrite III flattened anteriorly in males, lacking a distinct margin. Male Terminalia: Parameres (Fig. 7D) subparallel in basal 1/5, then concave and arcuately tapering to apex.

**Variation.** This species is fairly constant in its robust form. The main variation observed was in the elytral suture, which is usually entirely not elevated, but is occasionally produced in the posterior half, though is still very much shorter than the discal costae.

**Distribution.** Mojave Desert, from Edwards and Ridgecrest California, east through Mercury and Alamo, Nevada. This species is particularly abundant from sand dunes in the eastern Mojave and Death Valley (e.g., Kelso, Eureka, and Big Dune) (Fig. 9).

**Type material.** Holotype. "USA:CA: San Brndno/Co., Kelso Dunes; 34°/53′23″N, 115°43′04″W/April 16–17, 2011; at/night gleaning & UV/lights; W.B. Warner," "ARTSYS0007056," bearing red holotype label. Deposited in the ASUHIC, catalog number ASUHIC0101564. Paratypes: A total of 724 specimens from across its range, bearing blue paratype labels (see Data S1 and S2 or SCAN for full specimen data).

**Etymology.** This species is named for its robust stature (*Brown, 1956*) among *Trogloderus*.

**Remarks.** This species can often be recognized by gestalt, owing to its generally robust outline with a fusiform abdomen. One of the most abundant species in natural history collections, specimens were often previously been determined as *T. nevadus*. *T. major* is sympatric with *T. tuberculatus* and *T. vandykei*, where they are often taken in mixed series. This is the species from the Nevada Test Site referred to as *T. costatus nevadus* in *Tanner & Packham (1965)*, who reported this species active from March through October, with a distinct peak in abundance in August.

### *Trogloderus nevadus* La Rivers, 1943

Figures 6G, 7E and 9

**Diagnosis.** The combination of a punctate, evenly convex pronotum, tuberculate propleurae and epistoma, and the frontoclypeal suture forming a complete transverse ridge will separate this species from all congeners but some specimens of *T. arcanus*. See the key characters and diagnosis of the latter species to further separate the two.

**Redescription.** As genus with the following: Length 8.5–10 mm, width 3.5–4 mm. Head: Epistoma and frons tuberculate throughout, lacking distinct punctures; frontoclypeal suture forming complete transverse ridge. Thorax: Pronotum evenly convex dorsally; heavily punctate throughout, punctures longitudinally oval, tending to coalesce anteriorly; lateral margins moderately arcuate, sinuate in basal 1/5; posterior angles obliquely acute, small; anterior fovea weakly impressed, connected to posterior fovea; posterior fovea similarly weakly impressed, sometimes slightly deeper. Propleurae granulately tuberculate

throughout, always with tubercles present anteriorly underneath pronotal margin. Prosternal process horizontal, usually distinctly margined along entire outline, forming evenly tapering triangle behind posterior procoxal margin. Elytral costae moderately produced, intervals punctate, lacking transverse ridges; elytral suture weakly elevated in posterior 1/2. Abdomen: Abdominal depression indistinct to weak in females, discernable only on ventrite I, relatively weak in males, discernable on ventrites I–II, but lateral margin forming ridge only on ventrite I. Male terminalia: Parameres (Fig. 7E) triangular, evenly tapering from base to apex.

**Variation.** This species is fairly constant throughout its range. The pronotal foveae are sometimes moderately pronounced, generally in larger and more roughly sculptured individuals, whereas the typical form has the foveae very weakly depressed.

**Distribution.** This species is distributed throughout the northern Great Basin, throughout the Lake Lahontan drainage and into the Snake River Plains (Fig. 9).

**Type material.** Holotype male from Pyramid Lake Dunes, Washoe County, Nevada, not seen. Deposited in Ira La Rivers' collection (*La Rivers, 1943*: 439), which was later deposited at the state collection of Nevada in Reno, the type was not located there (K. Tonkel, 2018, personal communication), nor found at the CASC where a sizable amount of La Rivers material is located. The description, examined paratypes, and abundant subsequent collecting from the type locality leave no doubt as to this species identity.

**Material examined.** A total of 332 specimens including four paratypes (see Data S1 and S2 or SCAN for full specimen data).

**Remarks.** This species is broadly sympatric with *T. costatus*, but seemingly has a slightly broader range, extending south to the dunes around Walker Lake and north to Pyramid Lake. It is surprising that no specimens were found from southeastern Oregon, which seems to have appropriate habitat without any significant barriers to dispersal. Increased collecting efforts may produce specimens from the periphery of the currently known range. Many specimens referred to the present species in natural history collections belong to the herein described species with punctate pronota.

### *Trogloderus skillmani* Johnston, New Species

urn:lsid:zoobank.org:act:63C947D5-73A7-4188-A430-247B04AFD633

Figures 6D, 7F and 9

**Diagnosis.** This species can be recognized by the relatively evenly tuberculate pronotum, lack of subapical elytral tubercles, and relatively evenly tapering male parameres. This species is most similar to *T. verpus*, which can be separated by the male terminalia (parameres strongly constricted near base in *T. verpus*, parameres not strongly constricted, evenly tapering to apex in *T. skillmani*). The present species is also fairly similar to *T. warneri*, which can be separated by the pronotal characters given under the diagnosis for that species.

**Description.** As genus with the following: Length 9.5–12.5 mm, width 3.5–4 mm. Head: Epistoma and frons tuberculate throughout; mesal region of frons on same plane as clypeus; frontoclypeal suture not or weakly forming transverse ridge. Thorax: Pronotum relatively evenly convex dorsally; evenly tuberculate throughout, lateral regions of pronotum more or less depressed, but similarly tuberculate as remainder of disc; lateral margins fairly evenly arcuate, recurved just before posterior angles; posterior angles small, acute; posterior margin straight, mesal region forming continuous line laterally to terminus of posterior angle. Propleurae evenly and densely tuberculate throughout. Prosternal process short, usually offset dorsad from plane of prosternum between procoxae. Elytral costae moderately to strongly produced; intervals usually tuberculate, tubercles originating from center of interval as well as lateral faces of costae; elytral suture elevated in poster 3/4, nearly as produced as discal costae. Abdomen: Abdominal depression lacking in both sexes. Male parameres (Fig. 7F) narrowly triangular, evenly to slightly arcuately converging to apex.

**Variation.** This species as circumscribed here is the most widespread of any *Trogloderus* and has some significant variation accordingly. Specimens near the type locality, from northern Arizona and southern Utah, tend to have extremely tuberculate elytra intervals, strongly produced elytral costae, and small prosternal processes. Specimens from more typical great basin regions of Utah and Nevada (e.g., Little Sahara dunes, Crescent Dunes) tend to be less strongly sculptured on the elytra and have slightly enlarged prosternal processes. Specimens from the far western end of the distribution near Mono Lake have stronger elytral sculpturing and large, nearly horizontal prosternal processes. The posterior pronotal angles are always acute and usually form a continuous posterior margin to the pronotum, but occasionally the angles are obliquely oriented. This seems to be individual variation and not tied to geography.

**Distribution.** This species has the widest distribution of any *Trogloderus*, extending from the Coral Pink sand dunes and surrounding regions north to the Little Sahara Dunes and west to Mono Lake (Fig. 9).

**Type material.** Holotype: "USA: AZ: Mohave Co./6m E Colorado City/Rosy Canyon Road/1.5 m S UT state line/12-VII-2016/F.W. & S.A. Skillman," "ARTSYS0007053," bearing red holotype label. Deposited in the ASUHIC, catalog number ASUHIC0101565. Paratypes: A total of 920 specimens from the western regions of the Colorado Plateau around the Coral Pink Sand Dunes, Hurricane, and Toquerville Utah, bearing blue paratype labels (see Data S1 and S2 or SCAN for full specimen data).

**Other material.** A total of 182 specimens from the Northern and Western reaches of this species range.

**Etymology.** This species is named after Frederick W. Skillman, who both collected the holotype and has been a constant help throughout this study. His generous sharing of specimens, knowledge of natural history, and long drives to remote sand dunes are greatly appreciated.

**Remarks.** A broader molecular sampling and increased collections from Nevada localities may eventually find this taxon to be a cryptic species complex. Apparently able to cross boundaries that limit other species of *Trogloderus*, *T. skillmani* may be more adept at dispersing than its congeners.

### *Trogloderus tuberculatus* Blaisdell, 1909

= *Trogloderus costatus pappi* Kulzer, 1960

Figures 6A, 7G and 10

**Diagnosis.** This species can be readily identified by the presence of tubercles on the pronotum and the large, subapical tubercle at the terminus of the outer costa on each elytron. The present species can be further separated from the others with tuberculate pronota by the thick, raised ridges demarking the lateral margins and boundary between the pronotal foveae.

**Redescription.** As genus with the following: Length 10.5–12 mm, width 4–4.5 mm. Head: Epistoma and frons tuberculate throughout, lacking distinct punctures above antennal insertion; frontoclypeal suture forming complete, though gentle, transverse ridge; frontal tubercle covered with smaller tubercles. Thorax: Pronotum with dorsal silhouette appearing somewhat bilobed in anterior view; distinctly tuberculate throughout; lateral margins strongly arcuate, recurved just before posterior angles; posterior angles obliquely acute, small; foveae well demarked laterally by continuous strongly elevated longitudinal ridges; anterior fovea distinct, smooth, separated from posterior fovea by strongly elevated ridge; posterior fovea circular, usually smooth mesally. Propleurae fairly smooth, with dorsal longitudinal row of irregular tubercles running just beneath pronotal margin; often tuberculate on bulge covering procoxae. Prosternal process small, subtriangular, not margined laterally, slightly offset dorsad of prosternum between procoxae. Elytral costae strongly produced, crenulate; intervals with deep punctures, lacking transverse ridges; each elytron with subapical tubercle, formed by terminus of outer elytral carina, often formed by confluence of outer 1–3 costae; elytral suture very weakly produced, much shorter than discal carinae. Abdomen: Ventrites tuberculate throughout; abdominal depression weak, present on ventrites I–II in both sexes, without marginal ridge, usually somewhat smooth in males. Male terminalia: Parameres (Fig. 7G) narrowly triangle, more or less evenly tapering to apex.

**Variation.** The subapical elytral tubercles, unique to this species of *Trogloderus*, are somewhat variable. It is always made up of the thickened terminus of the outer elytral costa and is variably formed by the confluence of any combination of the outer three costae. This seems to be individual variation and not correlated with geography. Specimens from Kelso Dunes (the only confirmed locality where *T. tuberculatus* is sympatric with another species, *T. major*) are distinctly smaller than all other examined localities and possess less developed subterminal elytral tubercles.

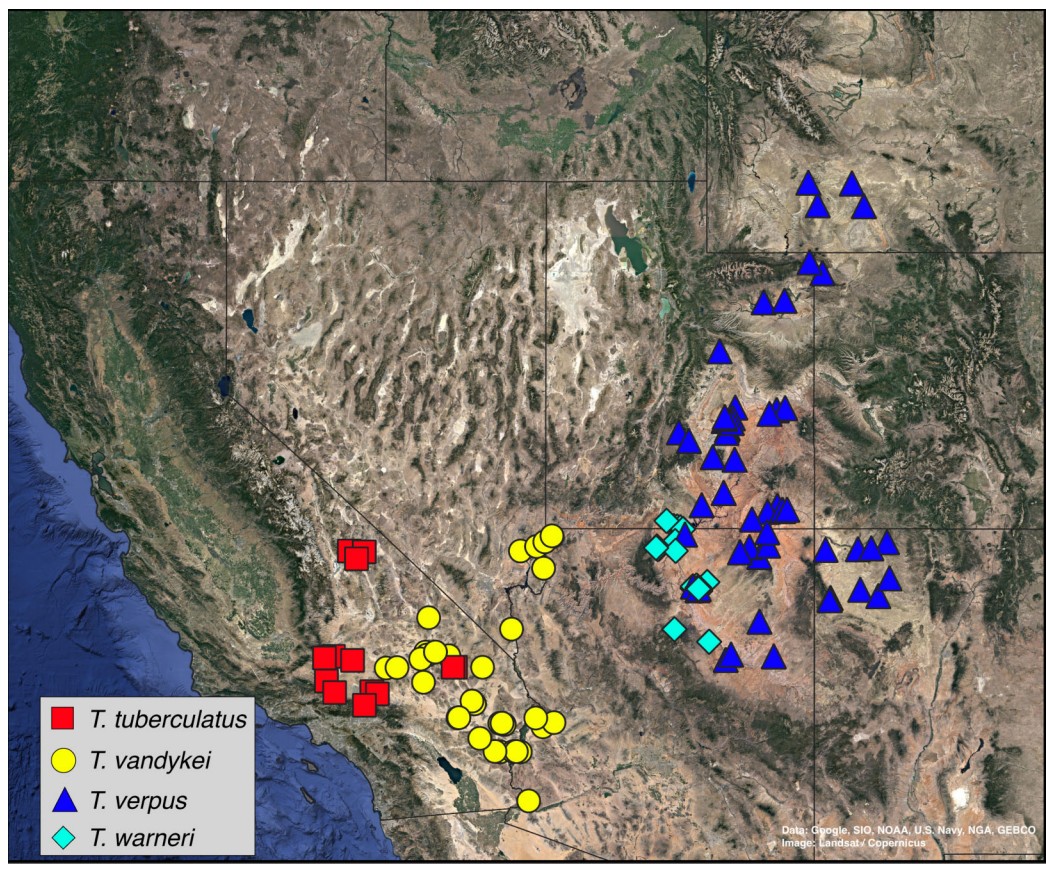

**Figure 10 Distribution map, *Trogloderus tuberculatus* Blaisdell, *T. vandykei* La Rivers, *T. verpus* Johnston n.sp., *T. warneri* Johnston n.sp.** Map data ©2019 Google, SIO, NOAA, U.S. Navy, NGA, GEBCO.

**Distribution.** This species is found in the Mojave Desert, and is generally found around the periphery from the western high desert reaches and in the northern Death Valley region (Fig. 10).

**Types.** The holotype of *T. tuberculatus* Blaisdell, collected from "L.A. County, California," was examined at the USNM. The holotype of *T. costatus pappi* Kulzer, from Lancaster, Mojave Desert, Southern California, was not examined (see remarks below).

**Material examined.** A total of 41 specimens (see Data S1 and S2 or SCAN for full specimen data).

**Remarks.** Similar to *T. costatus*, it is remarkable that this species, the least common in natural history collections, was the second species described in the genus. *Papp & Pierce (1960)* reported this species feeding on stored chicken feed in Lancaster, California. Specimens from this collecting event (the largest known for this species, at least ten individuals) were sent to the Frey museum in Germany (*Papp, 1961*: 35), which became the type series for *Trogloderus costatus pappi* Kulzer (*Kulzer, 1960*: 331). Though the type itself was not examined, seven specimens from *Papp's (1961)* original series were

studied and are all certainly conspecific with *T. tuberculatus* as herein circumscribed. This species was very difficult to recollect, particularly due to lack of suitable habitat. The western Mojave Desert has been largely developed, and after multiple targeted trips to the region only a single, ca. 0.25 acre, dune near California City was found to support a population of this species. Most specimens in natural history collections determined to this species (or subspecies as *T. costatus tuberculatus*) actually belong to other tuberculate species described herein.

### *Trogloderus vandykei* La Rivers, 1946

= *Trogloderus costatus mayhewi* Papp, 1961

Figures 1, 7H and 10

**Diagnosis.** This species can be readily separated from all other *Trogloderus* by the pronotal dorsum being bilobed when viewed from the front. Its pronotum is also punctate and more broadly explanate than any of its congeners. Most similar to and sympatric with *T. major*, the two can be readily separated by the given characters.

**Description.** As genus with the following: Length 9–11.5 mm, width 3.5–4.5 mm. Head: Epistoma punctato-tuberculate; frons smooth, mesal region on same plane as clypeus; frontoclypeal suture not forming complete ridge, obsolete at least mesally; frontal tubercle punctate, not very prominent. Thorax: Pronotal dorsum bilobed in anterior view; pronotum strongly explanate, punctate, punctured becoming irregular tubercles laterally; lateral margins strongly and evenly arcuate, recurved just before posterior angles; posterior angles obliquely acute, small; foveae bounded by raised lobed on either side, anterior fovea moderately impressed, forming continuous channel with posterior fovea, posterior fovea usually slightly deeper. Propleurae smooth, lacking tubercles throughout, occasionally with granulate tubercles ventrally around procoxae. Prosternal process horizontal, prominent, strongly margined, especially in males, forming evenly tapering triangle behind posterior procoxal margin. Elytral costae moderately produced, intervals relatively smooth, bering two rows of punctures, lacking any transverse ridges; elytral suture not at all elevated, situated in concavity formed by inner discal costae. Abdomen: Abdominal depression very strong in both sexes, exceedingly so in males, visible on ventrites I–III, demarked by strongly punctate lateral ridges which curve mesad and form distinct posterior margin on ventrite III. Male terminalia: Parameres (Fig. 7H) more or less arcuately converging from base to apex, apical 1/2 subparallel.

**Variation.** This species exhibits consistent morphology throughout its range. Occasionally smaller specimens are observed in which the pronotum appears less explanate, but this form seems sporadic, not tied to geography, and is likely a result of water or nutrient availability for the larva.

**Distribution.** Eastern and central Mojave Desert, especially abundant in dunes along the Colorado River (Fig. 10).

**Types.** The holotype of *T. costatus vandykei* La Rivers, from Baker, San Bernardino County, California, was examined at the CASC. The holotype of *T. costatus mayhewi* Papp, from Dale Dry Lake, San Bernardino County, California, was examined at the LACM.

**Material examined.** A total of 327 specimens (see Data S1 and S2 or SCAN for full specimen data).

**Remarks.** This species ranges the furthest south of any *Trogloderus* members, having been collected just north of Yuma, Arizona. *T. vandykei* has never been collected from the Algodones or other sand dunes in the Colorado Desert (*Johnston, Aalbu & Franz, 2018*). This is perhaps simply because they have not yet dispersed to these dunes. While there may be some other competitive or environmental factors at play, both *T. vandykei* and other congeners persist very well in regions of seemingly similar intense annual heat and dry conditions (e.g., Death Valley, Wiley's Well, Bouse Dunes, etc.).

### *Trogloderus verpus* Johnston, New Species

urn:lsid:zoobank.org:act:E413CFD5-4634-4321-8D85-03F9C8D85FEE

Figures 6B, 7I and 10

**Diagnosis.** This species can be recognized by the evenly tuberculate pronotum, lack of subapical elytral tubercles, and the male parameres being strongly constricted basally. It is most similar to *T. skillmani*, which can be separated by the male terminalia (parameres not constricted in *T. skillmani*).

**Description.** As genus with the following: Length 9.5–11.5 mm, width 3.5–4.5 mm. Head: Epistoma and frons evenly tuberculate, tubercles often irregularly shaped; frontoclypeal suture not forming complete transverse ridge, mesal region of frons more or less on same plane as clypeus. Thorax: Pronotum evenly convex dorsally, occasionally with lateral regions slightly flattened posteriorly; lateral marginsfairly evenly arcuate, recurved just before posterior angles; posterior angles small, acute; posterior margin usually straight, mesal region forming continuous line to terminus of posterior angle. Propleurae densely and evenly tuberculate throughout. Prosternal process acute, usually small, offset dorsad from plane of prosternum between procoxae. Elytral costae moderately to strongly developed, intervals variable from smooth to moderately tuberculate; elytral suture moderately to strongly elevated in posterior ½, usually distinctly shorter than discal costae. Abdomen: Abdominal depression absent in both sexes. Male Terminalia: Parameres (Fig. 7I) strongly constricted near basal 1/6, then narrowly and evenly tapered to apex.

**Variation.** This species is fairly consistent across its range, but presents some variation in the elytral sculpturing. In some specimens the intervals between discal costae are noticeably tuberculate, while most are smooth. The elytral suture is usually less strongly elevated than the discal costae, but in some New Mexico populations (e.g., near Farmington), it is nearly the same height as the discal costae. Specimens from the sand dunes near Moenkopi, where they are sympatric with *T. warneri*, are distinctly smaller and less roughly sculptured than anywhere else in its range.

**Distribution.** This species is broadly distributed throughout the Colorado Plateau, from Moenkopi, Arizona east to central New Mexico and north to the Killpecker Dunes in Wyoming (Fig. 10).

**Type material.** Holotype: "USA: UT: Grand Co./22m NW Moab, Dubinky/Well Rd. @ Dubinky Well/25-VI-2016/Skillman & Johnston," "ARTSYS0007055," bearing red holotype label. Deposited in the ASUHIC, catalog number ASUHIC0101566. Paratypes. A total of 185 specimens from throughout the species range bearing blue paratype labels (see Data S1 and S2 or SCAN for full specimen data).

**Etymology.** This species name is given for the strongly constricted male parameres, which look as though a portion has been cut away from the fairly regularly triangular shape found in the rest of the genus (*Brown, 1956*).

**Remarks.** The remarkably small specimens from near Moenkopi may be an example of competition forcing allometry. Indeed, the specimens of *T. warneri* from Moenkopi are a very similar size to specimens of *T. verpus* from the rest of its range.

### *Trogloderus warneri* Johnston, New Species

urn:lsid:zoobank.org:act:9D1E0BF2-F309-4A98-A64C-5708CFE864D0

Figures 6C, 7J and 10

**Diagnosis.** This species can be recognized by the combination of a tuberculate pronotum and large, inflated posterior pronotal angles. The species can be further recognized by the depressed lateral regions of the pronotum lacking tubercles, the lack of an abdominal impression, and the lack of subapical elytral tubercles.

**Description.** As genus with the following: Length 9–11 mm, width 3.5–5 mm. Head: Epistoma and frons tuberculate throughout, frontoclypeal suture not developed as transverse ridge, mesal region of frons on same plane as clypeus. Thorax: Pronotum relatively evenly conxex dorsally; heavily tuberculate; disc laterally depressed, usually lacking tubercles, especially posteriorly; lateral margins arcuate, more strongly narrowed posteriorly, recurved just before posterior angles; posterior angles large, obliquely angles, usually well inflated, sometimes broadly acute. Propleurae evenly tuberculate, tubercles fairly large and rounded. Prosternal process short, triangular, offset dorsad from plane of prosternum between procoxae. Elytral costae well developed, intervals with deep punctures, sometimes giving appearance of short transverse ridges; elytral suture weakly to moderately produced in posterior half, always shorter than discal costae. Abdomen: Ventrites tuberculate; without abdominal depression, ventrite I sometimes smooth mesally in males. Male terminalia: Parameres (Fig. 7J) subparallel in basal 1/5, then arcuately converging to apex.

**Variation.** The pronotum, while diagnostic for this species, is somewhat variable in the specimens examined. The typical form has very strongly inflated posterior angles and the disc distinctly depressed and lacking tubercles laterally. In some specimens the posterior

angles are less inflated and the depressed lateral region is much smaller, tending to be restricted to the posterior third. However, these reduced characters were distinctly discernable in all specimens studied, reliably separating them from other *Trogloderus* species.

**Distribution.** Distributed in the western Colorado Plateau, the species seems bounded on the west by the Kaibab Plateau, and are distributed as far east as Moenkopi, Arizona (Fig. 10).

**Type material.** Holotype: "USA:AZ:Coconino Co./Hwy. 264 2.2mi SE jct/US160; 36°05′57″N/111°12′03″W; dunes at/night; April 20, 2012; W.B. Warner, J.P. Gruber." "ARTSYS0007054," bearing red holotype label. Deposited in the ASUHIC, catalog number ASUHIC0101563. Paratypes: A total of 237 specimens from across the species range (see Data S1 and S2 or SCAN for full specimen data).

**Etymology.** I am honored and thankful to name this species for William B. Warner, an ardent collector, coleopterist, and natural historian. His assistance and encouragement throughout this project in both the field and the lab are greatly appreciated.

**Remarks.** This species has a relatively small geographic range, yet extends across the eastern reaches of the Grand Canyon. This is perhaps the reason for the observed moderate genetic diversity within the species. Most specimens in natural history collections have been determined as *T. tuberculatus*.

# RESULTS

## Phylogenetic reconstruction

Both maximum likelihood and Bayesian analyses converged on a single topology with moderately strong support throughout (Fig. 11). Within the outgroups, the genus *Eleodes* Eschscholtz was notably recovered as paraphyletic with respect to the genera *Neobaphion* Blaisdell, *Embaphion* Say, *Lariversius* Blaisdell, and *Trogloderus*. This raises broader questions regarding the naturalness of the current classification of the tribal concept as a whole (*Bousquet et al., 2018*); however, the sampling for this study is not sufficient to justify more substantive classificatory changes. *Trogloderus* was recovered as monophyletic, and is further subdivided into two strongly supported clades—that is, (1) the "tuberculate-pronotum clade" containing all species that bear distinct tubercles on the pronotal disc, and (2) the "reticulate-pronotum clade" containing all species whose pronotal discs have deep punctures that make the intervals appear to be elevated into reticulate sculpturing. All *Trogloderus* species as circumscribed above were similarly found to be monophyletic with posterior probabilities of 1 and bootstrap support of 95 or higher.

The *Trogloderus* tuberculate-pronotum clade contains four species and is well resolved (Fig. 11), with internal nodes between species all having posterior probabilities greater than 0.95 and bootstrap values above 85. The relationships between these species imply an east-to-west diversification pattern. The easternmost species, *T. verpus* (Fig. 10) known from the Colorado Plateau, is recovered as sister to a clade containing the remaining three

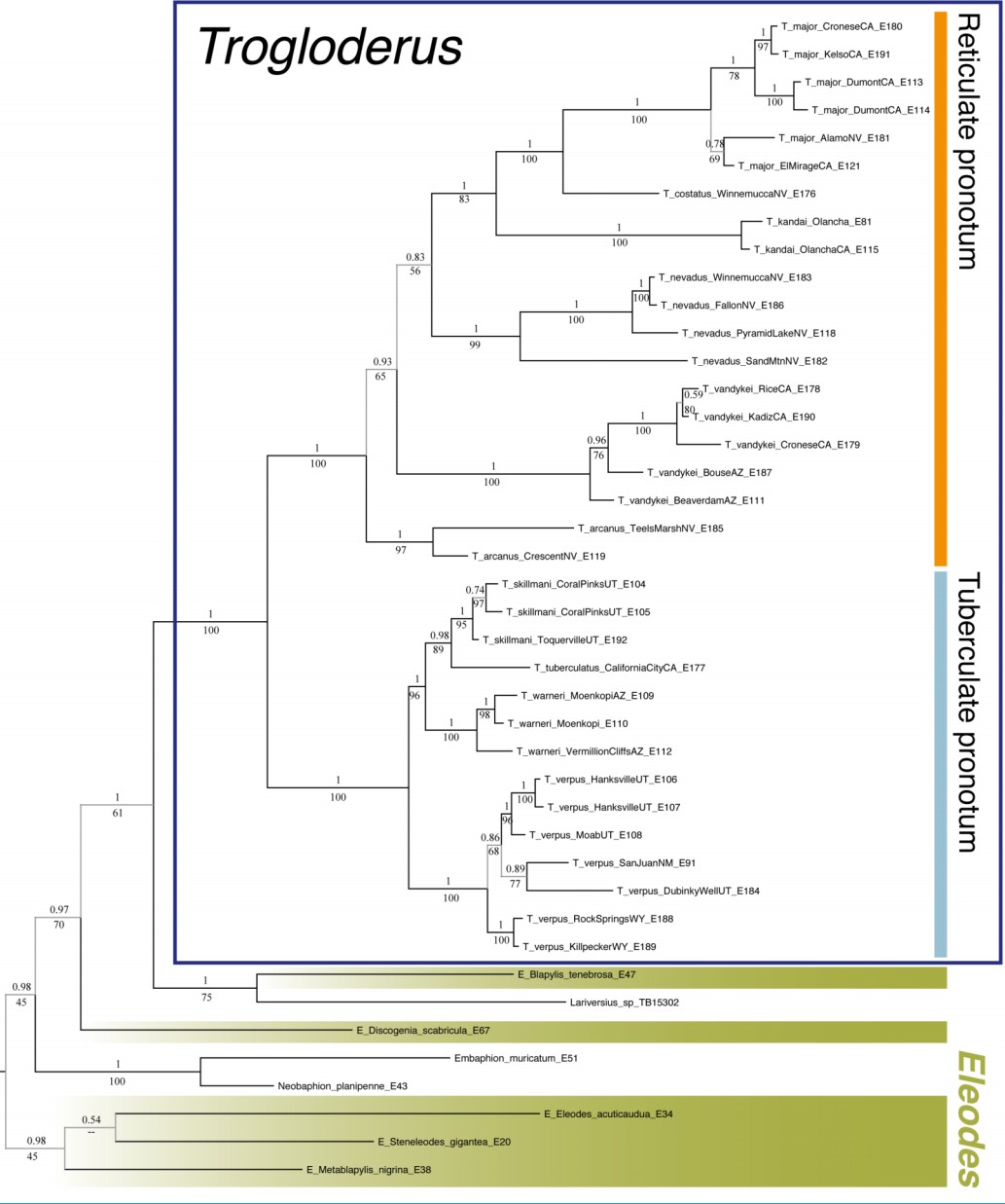

**Figure 11 Phylogenetic reconstruction of *Trogloderus*.** Tree shown is from the MrBayes analysis, numbers above branches are posterior probabilities, numbers below the branches arethe corresponding RAxML bootstrap support values. Outgroup specimens belonging to the genus *Eleodes* are highlighted. The monophyletic *Trogloderus* is indicated by a box, and the reciprocally monophyletic Reticulate-pronotum and Tuberculate-pronotum clades are indicated by vertical bars.

species. The latter clade shows the same trend with its easternmost species, *T. warneri* (Fig. 10) distributed east of the Kaibab Plateau, sister to the species *T. skillmani* (Fig. 9) and *T. tuberculatus* (Fig. 10), which are distributed west of the Kaibab Plateau.

The *Trogloderus* reticulate-pronotum clade contains six species (Fig. 11) with notably western distributions, ranging from the Mojave Desert to the Great Basin.

The relationships between these species are less well resolved than for those of the tuberculate-pronotum clade, though each species is supported as monophyletic with posterior probabilities of 1 and bootstrap support values of 95 or higher. While analyses converged on a single topology, the underlying data do not give unequivocal support to the relationships of the early-diverging species. *Trogloderus arcanus*, *T. vandykei*, and *T. nevadus* are inferred to have diverged before a clade containing the other three reticulate-pronotum species. However, these branches all have posterior probabilities lower than 0.95 and bootstrap support values below 75. The clade consisting of *T. kandai*, *T. costatus*, and *T. major* is strongly supported with a posterior probability of 1 and a bootstrap support value of 83. The reticulate-pronotum clade seems to indicate a latitudinal pattern to diversification. Neither of the two sympatric pairs of species in this clade, the southern *T. vandykei* with *T. major* and the northern *T. costatus* with T. *nevadus*, form monophyletic groups. This supports the notion that multiple vicariant or dispersal events between these regions were involved in the diversification of this lineage.

*Trogloderus arcanus* and *T. nevadus* exhibit longer branch lengths between sampled populations within the species than any others sampled for this study (Fig. 11). This may simply be due to limited sampling, but further molecular and morphological investigations from the under-sampled regions of Nevada may provide evidence for the two herein circumscribed species to represent more complex taxonomic groups.

## Phylogenetic dating analyses

*Trogloderus* is here inferred to be relatively young, with the most recent common ancestor (MRCA) for the genus occurring during the late Miocene or earliest Pliocene (Fig. 12). Furthermore, most speciation events are inferred to have taken place during the Pleistocene. Based on these inferences, it seems evident that *La Rivers (1946)* hypothesis of an ancient lineage approaching extinction can be refuted for *Trogloderus*. Instead, *Trogloderus* seems to postdate the Neogene Uplift, having originated and diversified in conjunction with the recent desert formations of western North America (*Wilson & Pitts, 2010*).

Dating analyses for *Trogloderus* using BEAST (Figs. 12A and 12B) inferred comparatively older dates than RelTime (Figs. 12C and 12D) but are not particularly reliable, having failed to converge after 500 million generations. The MRCA of *Trogloderus* was dated to 10.27 mya, and ages for both calibrated nodes were older than expected, namely 4.03 mya for the Inyo-White mountains calibration, with the prior mean set at 2.5 mya, and 1 mya for the Grand Canyon calibration, with a prior mean set at 0.83 mya. The estimated sample sizes for mutation rates did not exceed 10 and those for calibration times and tree height were well under 100. Additional analyses under different locus partition and model schemes and modified taxon inclusion similarly failed to converge. This may be due either to limitations with the underlying molecular dataset, or because the coalescent-based priors may be inappropriate for this class of data. The results using the Yule model are shown in Figs. 12A and 12B, displaying the median node age and 95% highest posterior density respectively. Due to this lack of convergence, the timetree from RelTime was used for subsequent historical biogeographic inference.

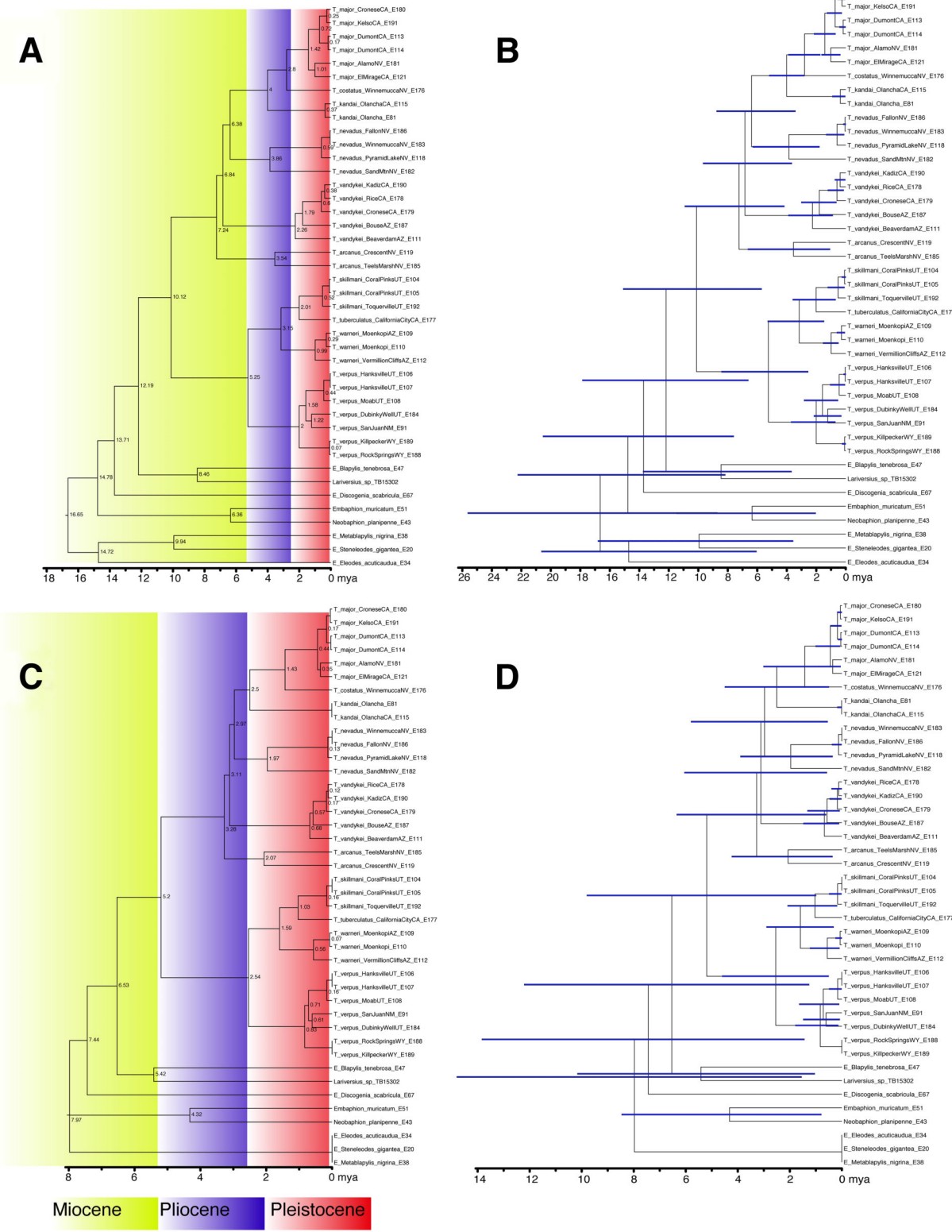

**Figure 12 Diversification estimates for *Trogloderus*.** (A) Timetree generated from BEAST showing inferred median node ages. (B) Same showing 95% highest posterior density for node ages. (C) Timetree generated from RelTime showing inferred median node ages. (D) Same showing 95% confidence intervals for node ages.

RelTime analyses infer *Trogloderus* to have originated in the earliest Pliocene with most current species arising during the mid-Pleistocene. Divergence estimates from RelTime were consistently later than those inferred from BEAST, with the MRCA of *Trogloderus* dated to 5.2 mya, and the dates of 2.5 and 0.56 mya for the calibration clades split by the Inyo-White mountains and the Grand Canyon respectively. Median node ages and 95% confidence intervals inferred from RelTime are shown in Figs. 12C and 12D respectively.

The Sierra Nevada mountains offer one line of geological evidence for the age of *Trogloderus* to be closer to 5 my as the RelTime analysis infers. The timing of the uplift of the Sierra Nevada Mountains remains contested in the geological literature (*Wilson & Pitts, 2010*), but significant evidence suggests that the majority of the uplift occurred between 5 and 8 mya and was a primary force in creating the Great Basin and Mojave deserts (*Jones, Farmer & Unruh, 2004*; *Wilson & Pitts, 2010*). Were *Trogloderus* older than this uplift event, we might expect them to be present outside of the intermountain region. Indeed, members of the genus are able to endure cold winters from central Wyoming as well as the extreme heat from Death Valley and surrounding environs. Beyond living in sandy substrates, there are no other clear environmental limits to their distribution.

The MRCA of all included Amphidorini taxa was dated to 7.97 mya with a 95% confidence interval of 1.5–14.5 mya using RelTime. This date range, though the first inferred for this fossil-lacking tribe, is younger than expected based on phylogenetic work at the family level. The new-world Amphidorini appear to be sister the old-world tribe Blaptini Leach, 1815 (*Kanda, 2017*). The latter was estimated by *Kergoat et al. (2014)* to have an origin closer to 55 mya, but no members of Amphidorini were included in that study. The young age inferred here for the tribe may again be a symptom of low species-level taxon sampling. Hypotheses about the origin and diversification of Amphidorini will have to wait for future studies with a broader scope.

### Historical biogeographic estimation

The MRCA of *Trogloderus* was inferred to inhabit the Colorado Plateau (Fig. 13), where the majority of the tuberculate-pronotum clade still resides. The ancestors of the reticulate-pronotum clade are inferred to have dispersed into the Lahonton Trough, and from there radiated into the Mojave Desert, Great Basin, and Owens Valley. Three separate radiations into the Mojave Desert are inferred for the three species sympatric there. The insights given by this biogeographic estimation for specific subregions are discussed in detail below.

Historical biogeographic estimation in BioGeoBEARS supports the use of a model incorporating founder-event jump dispersal (*Matzke, 2014*). This process is not only important for taxa distributed across islands (*Matzke, 2014*; *Zhang et al., 2017*), but also for taxa living on sand dunes or other isolated habitats which can functionally act the same as islands (*Van Dam & Matzke, 2016*). The DEC model resulted in a most likely estimation with a log likelihood score of −43.6. The DEC+J model, which employs a single
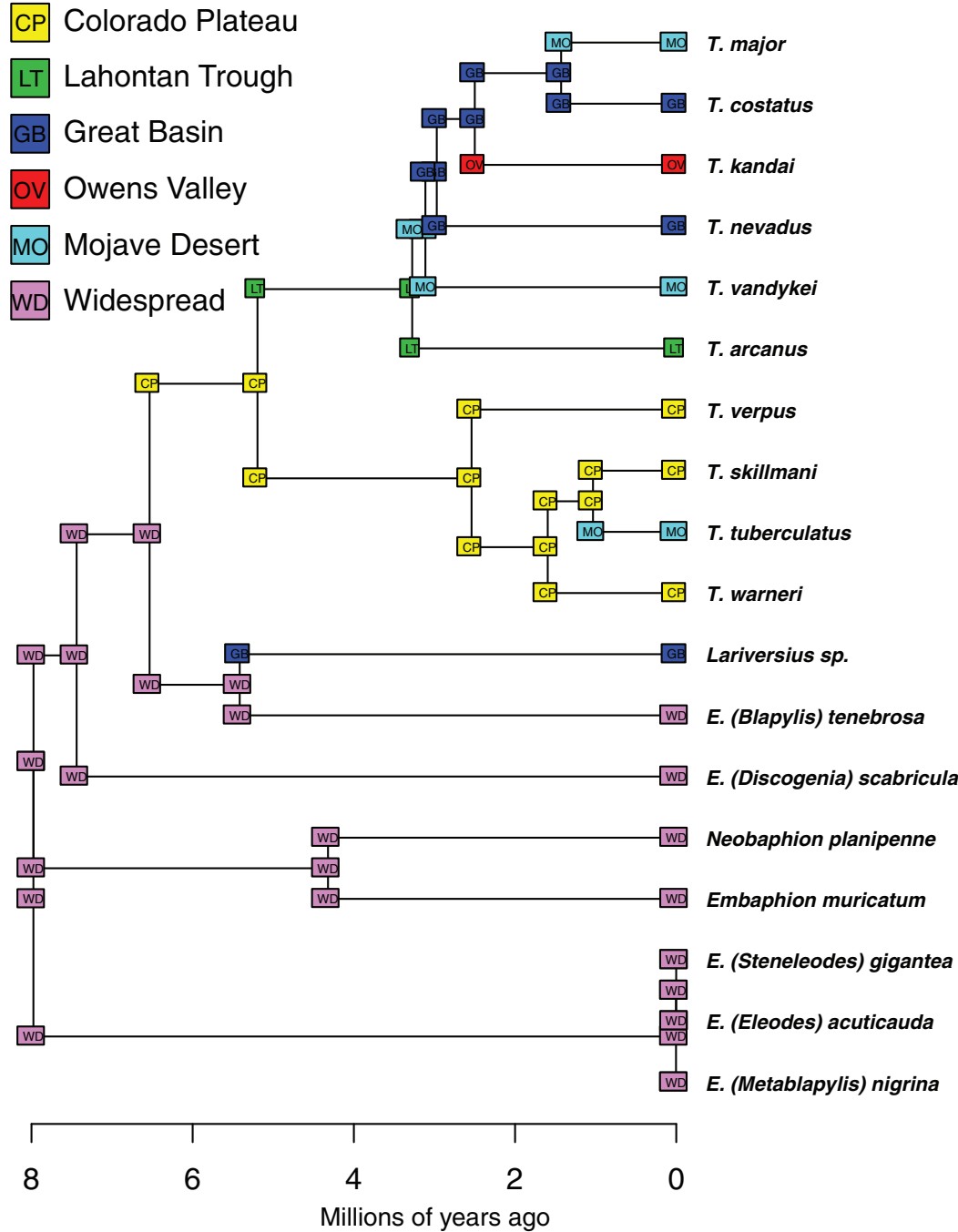

**Figure 13 Historical biogeographic estimation of *Trogloderus*.** Generated from BioGeoBEARS using the DEC+J model. Nodes colored by inferred most likely biogeographic region.

extra parameter for jump dispersal, produced an estimation with a log likelihood of −26.8. By performing a likelihood ratio test (*Huelsenbeck & Crandall, 1997*), the DEC+J model provides a significantly better fit to the data than the DEC model at a *P*-value of 1e−5.
## Discussion of the biogeography of the Intermountain Region

The historical biogeography of *Trogloderus* supports the distinction of the Lahontan Trough as a unique element of the Intermountain Region, and is the first to provide molecular and historical biogeographic support for the area to play a part in the migration of clades throughout the intermountain region. The appraisal of the biogeography of the Intermountain Region by *Reveal (1979)* was a landmark study based largely on floristic distributions and extensive field observations. A comprehensive biogeographic review of the region has not been published since. One major hypothesis put forth in this work is that the Lahontan Trough acts as a migration route into and out of the region. Following the establishment of the Lahontan Trough as a biogeographic entity by *Reveal (1979)*, multiple studies have found populations from this area to be distinct from populations of the same species from the Mojave and Great Basin deserts (*Britten & Rust, 1996*; *Hafner, Reddington & Craig, 2006*), and at least one psammophilic plant is unique to the area (*Pavlik, 1989*). Together, these studies suggest that the Lahontan Trough is likely to play an important role in the evolutionary history of any sand-dune restricted or dispersal-limited taxa in the region.

The newly described *T. kandai* is the first sand-dune species known to be restricted to the southern Owens Valley. The region has been relatively well studied for changes in plant communities (*Koehler & Anderson, 1995*; *Elmore, Mustard & Manning, 2003*) and fish conservation (*Galicia et al., 2015*). However, the sand dunes, which are comprised of particles originating from the surrounding Sierra Nevada and Coso mountains (*Lancaster et al., 2015*), have not had any beetle species reported only from them (*Andrews, Hardy & Giuliani, 1979*). Whether *T. kandai* is truly the only species restricted to this habitat or if there are others waiting to be described, additional faunal surveys of the sand dunes around the dry Owens Lake should be completed to understand what further importance this area may have for Intermountain biodiversity.

The three sympatric species of *Trogloderus* with independent dispersal events into the Mojave Desert are consistent with the inference of an eastern origin for the genus with a continual movement westward. The relatively recent timing for incursions into the Mojave Desert is also consistent with the fact that *Trogloderus* does not range south into the dunes of the Colorado and Sonoran deserts (*Aalbu & Smith, 2014*; *Johnston, Aalbu & Franz, 2018*). The relationships of the dune systems within the Mojave Desert were subdivided and well-tested by *Van Dam & Matzke (2016)*, but are here treated as a single unit. The barriers between these sand systems within this area seem to not be a major limiting factor for *Trogloderus* as *T. tuberculatus* and *T. major* are fairly evenly spread throughout.

The predicted footprint of the prehistoric lakes making up the Bouse Embayment is almost identical to the distribution of *T. vandykei*. This region, spanning along the lower Colorado River between Arizona and California (*Wilson & Pitts, 2010*), was covered by three large prehistoric lakes that ran from just north of present-day Bullhead City, Arizona south past Blythe, Arizona. The drainage was bounded along the south by the Chocolate Mountains and extended west into the Bristol basin (*Spencer et al., 2013*).

These lakes likely appeared around 4.9 mya and drained relatively shortly thereafter when the Colorado River eventually connected to the Gulf of California (*Spencer et al., 2013*). It is very likely that the sand derived from these lakes and the geological boundaries that formed their drainage basins have shaped the diversification and distribution of *T. vandykei*. The lakes are also implicated in genetically structuring the populations of a desert scorpion (*Graham et al., 2017*). The Bouse Embayment is further supported as a separate biogeographic entity based on the distribution of other psammophilic Tenebrionidae. Though the Algodones dunes are in extremely close proximity to the southern edge of the Bouse Formation, not only does *Trogloderus* not cross over the Chocolate Mountains and occur there, but multiple species restricted to the Algodones and Gran Desierto de Altar similarly do not extend north into the Bouse Embayment (*Johnston, Aalbu & Franz, 2018*).

Within the Colorado Plateau, three subregions are suggested by *Trogloderus* distributions. The distribution of the eastern *T. verpus* is somewhat surprising in that no previous biogeographic hypotheses were found to explain why it does not range as far west as the Vermillion Cliffs. One explanation is competitive exclusion within the genus, and this is somewhat supported by the populations near Moenkopi, Arizona. Both *T. verpus* and *T. warneri* occur on these dunes, and all studied specimens of *T. verpus* were significantly smaller than those of *T. warneri*. However, throughout the rest of its range, *T. verpus* has roughly the same body size as *T. warneri*. Another possible explanation is that the Kaiparowitz Formation around Grand Staircase-Escalante National Monument acts as a barrier between sand systems from the Kaibito and Moenkopi plateaus of north-central Arizona and those from the northern reaches of the greater Colorado Plateau. The Kaiparowitz Formation, along with the Wasatch Mountains, formed the western boundary of the western interior seaway during the Cretaceous (*Hettinger et al., 1996*; *Roberts, 2007*) and is implicated in the speciation of large dinosaurs at the time (*Sampson et al., 2010*). No studies of modern taxa that study this boundary were found. Even though the Colorado River and its tributaries have carved large canyons through this formation, it may still be a significant barrier between sand-dune restricted taxa. The third subregion is separated from the others by the Kaibab Plateau. This tall formation separates *T. warneri* from its eastern *T. skillmani* and *T. tuberculatus*. The effect of the Kaibab Plateau on dune-dwelling taxa is apparently similarly unstudied.

## CONCLUSIONS

The revision and historical biogeography of *Trogloderus* help to bring the biogeographic trends of the Intermountain Region into focus. The cohesive distributional patterns of *Trogloderus* species build upon the foundational work of *Reveal (1979)* and highlight regions that should be critically evaluated during future phylogenetic, taxonomic, and biogeographic studies. It is hoped that continued research on the under-studied biodiversity of the Intermountain Region will continue to bring clarity to the relationships between sand-dune systems of western North America.

## ACKNOWLEDGEMENTS

The author thanks Aaron Smith and Kojun Kanda for their sharing of tenebrionid knowledge and assistance with molecular analyses. Nico Franz supported this study from conception to completion and offered valuable feedback on the manuscript. William Warner and Frederick Skillman are gratefully acknowledged for their support during field work and along with many curators and managers of natural history collections provided the necessary specimens for this study. Comments and suggestions from three reviewers provided improvements to earlier versions of this manuscript.

### Funding

This research was supported by the National Sciences Foundation (DEB–1258154 and DEB–1754731), the United States Department of Agriculture—Agricultural Research Service (Agreement 58–1275–1–335), the CanaColl Foundation, and the Canadian Museum of Nature visiting scientist program. The funders had no role in study design, data collection and analysis, decision to publish, or preparation of the manuscript.

### Grant Disclosures

The following grant information was disclosed by the authors:
National Sciences Foundation: DEB–1258154 and DEB–1754731.
United States Department of Agriculture—Agricultural Research Service: Agreement 58–1275–1–335.
CanaColl Foundation, and the Canadian Museum of Nature visiting scientist program.

### Competing Interests

The author declares that they have no competing interests.

### Author Contributions

- M. Andrew Johnston conceived and designed the experiments, performed the experiments, analyzed the data, contributed reagents/materials/analysis tools, prepared figures and/or tables, authored or reviewed drafts of the paper, approved the final draft.

### DNA Deposition

The following information was supplied regarding the deposition of DNA sequences:
The DNA sequences are available as a Supplemental File and at GenBank: MN597189–MN597394.

### Data Availability

Full specimen data are available on the Symbiota Collections of Arthropods Network: http://scan-bugs.org and also are available in the Data S1 and S2.

## New Species Registration

The following information was supplied regarding the registration of a newly described species:

Publication LSID:

urn:lsid:zoobank.org:pub:678EBFE3-6308-4FB8-8E93-184CEC9A15E7.

*Trogloderus arcanus* Johnston LSID:

urn:lsid:zoobank.org:act:0BCDA9E8-F615-41B9-9376-62778B0958EE.

*Trogloderus kandai* Johnston LSID:

urn:lsid:zoobank.org:act:09FCBD7E-3DE8-40E4-8C13-5EC6DFA734EA.

*Trogloderus major* Johnston LSID:

urn:lsid:zoobank.org:act:1B61B89E-5839-47CA-AA20-F4795FF931D7.

*Trogloderus skillmani* Johnston LSID:

urn:lsid:zoobank.org:act:63C947D5-73A7-4188-A430-247B04AFD633.

*Trogloderus verpus* Johnston LSID:

urn:lsid:zoobank.org:act:E413CFD5-4634-4321-8D85-03F9C8D85FEE.

*Trogloderus warneri* Johnston LSID:

urn:lsid:zoobank.org:act:9D1E0BF2-F309-4A98-A64C-5708CFE864D0.

## Supplemental Information

Supplemental information for this article can be found online at http://dx.doi.org/10.7717/peerj.8039#supplemental-information.

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
