# Peer review of "Phylogenetic revision of the psammophilic Trogloderus LeConte (Coleoptera: Tenebrionidae), with biogeographic implications for the Intermountain Region"

_PeerJ, doi:10.7717/peerj.8039_

## Round 0.1 · original submission · Major Revisions

Your paper has now been carefully read by three reviewers. All three considered the manuscript to be a solid, well written contribution. I likewise find it to be an excellent paper with a wide breadth of analyses and data exploration; the descriptions are thorough and VERY nicely illustrated. That said, Reviewer 2 points out a number of methodological issues that will need to be addressed in a revised manuscript - criticisms if addressed mostly likely require some reanalysis of the data (albeit not overly extensive). As such I am returning the manuscript back to you with the recommendation that it requires major revisions (in all fairness it's more somewhere in between minor and major).

When you revise your paper, I would like to ask that you pay careful attention to all of the comments, both in the review narratives and in the two marked up manuscript drafts. In your cover letter please document explicitly how you addressed each of the comments provided by all three reviewers.

This manuscript was a pleasure to read; I look forward to seeing a revised copy.

·

Basic reporting

I thought that the article was well-written, thoroughly researched and included all aspects of a well thought-out manuscript. It includes important taxonomic, phylogenetic and biogeographic data. I typically provide a large number of comments in my reviews but this manuscript requires only a small number of minor comments (included as comments in the pdf file attached).

Experimental design

The goals for this study are clear and the results based on up-to-date analytical methods. I like the paragraph on nomenclatural acts in electronic only journals.

Validity of the findings

The taxonomic, phylegenetic and biogeographic data are significant and useful to tell the evolutionary story of this genus of darkling beetle. The conclusions are well thought-out and the results will serve as very good hypotheses to test when supplementary data is available.

Additional comments

A well-written manuscript, looking forward to see it published.

Reviewer 2 ·

Basic reporting

Recommendation: Accept with major revisions

Comments:

Do you wish to remain anonymous?: Yes

Is the author aware of the background and source material to the problems set forth?: Yes

Are the conclusions justified by the evidence presented and the assumptions involved?: Yes

Are the illustrations and tables clear and understandable?: Yes

In number are they: Sufficient


This is a very interesting paper on a largely understudied biogeographic region that revises a genus and investigates its historical biogeography. It is well written and concise which is appreciated. In general, it is a very sound study and it is clear that the author is a real authority on the study organism. I only had two issues with the paper, once these and other comments are addressed it should be ready for final acceptance and publication: First, that the biogeographic analyses should not include multiple tips per-species as this violates some of the model assumptions in BioGeoBEARS. It will likely not change much when reanalyzed as all of the species are found in a single area but you will probably want your analyses to be sound. Second, I would suggest dropping the BEAST analyses or do a full investigation as to what is going on in the data to try and see if there are any parameters to be adjusted. Please see my comments below in the paper on this. Your other divergence date estimates are largely in concordance and overlapping with the geological processes of interest, so I am not sure what is gained by presenting the BEAST analyses that did not converge.

Smaller things: Consider taking pictures of the TYPE labels rather than trying to transcribe verbatim. Also consider designating a new TYPE for T. nevadus as it seems to be lost.


See specific comments in attached word doc.

Experimental design

see word doc for specifics

Validity of the findings

see word doc for specifics

Additional comments

see word doc for specifics

Annotated reviews are not available for download in order to protect the identity of reviewers who chose to remain anonymous.

Reviewer 3 ·

Basic reporting

Some slight asymmetry or empty space in morphological and habitus figures.

Experimental design

no comment

Validity of the findings

Within my knowledge the applicable data are provided and conclusions sound.

Additional comments

Line 682: Italicize “T. kandai” and please check for consistency throughout.

Line 745: Italicize “T. arcanus” and please check for consistency throughout.

Line 1002: Italicize “T. verpus” and please check for consistency throughout.

General note on morphological and habitus figures: A scale bar in each figure or each constituent element might be of use to readers; please consider adding, if practical.

Figure 3: Improve the cohesiveness of mouthparts C-E by better blending the overall gray background with off-gray background near each disarticulated mouthpart. A portion of D is overlapping B.

Figure 4. The use of both low-contrast reds and greens is likely to present issues for colorblind readers. Please consider an alternate highlight color.

Figure 6. Habitus E appears too dark to discern features; please lighten. Consider lightening C, G, and H too as they may appear darker on other computer monitors.

---

## Round 0.2 · accepted · Accept

Thanks for the careful attention to reviewers' comments!